# Exploring therapeutic strategies for infantile neuronal axonal dystrophy (INAD/PARK14)

Guang Lin[1,2], Burak Tepe[1,2], Geoff McGrane[3], Regine C Tipon[3], Gist Croft[3], Leena Panwala[4], Amanda Hope[4], Agnes JH Liang[1,2], Zhongyuan Zuo[1,2], Seul Kee Byeon[5], Lily Wang[1,2], Akhilesh Pandey[5,6], Hugo J Bellen[1,2,7]*

[1]Department of Molecular and Human Genetics, Baylor College of Medicine, Houston, United States; [2]Jan and Dan Duncan Neurological Research Institute, Texas Children's Hospital, Houston, United States; [3]New York Stem Cell Foundation Research Institute, New York, United States; [4]INADcure Foundation, Jersey City, United States; [5]Department of Laboratory Medicine and Pathology, Mayo Clinic, Rochester, United States; [6]Manipal Academy of Higher Education, Manipal, Karnataka, India; [7]Department of Neuroscience, Baylor College of Medicine, Houston, United States

**Abstract** Infantile neuroaxonal dystrophy (INAD) is caused by recessive variants in *PLA2G6* and is a lethal pediatric neurodegenerative disorder. Loss of the *Drosophila* homolog of *PLA2G6*, leads to ceramide accumulation, lysosome expansion, and mitochondrial defects. Here, we report that retromer function, ceramide metabolism, the endolysosomal pathway, and mitochondrial morphology are affected in INAD patient-derived neurons. We show that in INAD mouse models, the same features are affected in Purkinje cells, arguing that the neuropathological mechanisms are evolutionary conserved and that these features can be used as biomarkers. We tested 20 drugs that target these pathways and found that Ambroxol, Desipramine, Azoramide, and Genistein alleviate neurodegenerative phenotypes in INAD flies and INAD patient-derived neural progenitor cells. We also develop an AAV-based gene therapy approach that delays neurodegeneration and prolongs lifespan in an INAD mouse model.

*For correspondence: hbellen@bcm.edu

## Editor's evaluation

This important study is of significant interest to those studying neurodegeneration, demonstrating key pathologies in PLA2G6-associated disease in both patient-derived neuronal models and a novel trans heterozygote mouse model. It identifies a number of possible compounds that could potentially be re-purposed for therapeutic use in PLA2G6-associated neurodegeneration and provides a proof-of-principle in mouse that gene therapy with human PLA2G6 can rescue defects in PLA2G6 deficiency. The data are solid and convincing, and will stimulate future work to elucidate the precise molecular mechanisms at play.

## Introduction

Infantile neuroaxonal dystrophy (INAD) (OMIM #256600) is a devastating and lethal pediatric neurodegenerative disorder caused by recessive variants in *PLA2G6* (*Khateeb et al., 2006*; *Morgan et al., 2006*). Moreover, variants in *PLA2G6* also cause atypical NAD (aNAD) (OMIM # 610217) and *PLA2G6*-related dystonia-parkinsonism, also called Parkinson disease 14 (PARK14) (OMIM #612953)

(*Paisan-Ruiz et al., 2009*). Collectively, these three diseases are called *PLA2G6*-associated neurodegeneration (PLAN) (*Kurian et al., 2008*). Iron accumulation has been observed in the basal ganglia of some INAD and aNAD as well as PARK14 patients (*Ferese et al., 2018*; *Yoshino et al., 2010*). Hence, these diseases are also categorized as neurodegeneration with brain iron accumulation 2 (NBIA2) (OMIM #610217) (*Gregory et al., 2008*; *Morgan et al., 2006*). The symptoms of INAD include early onset ataxia (age 1–3 years), mental and motor deterioration, hypotonia, progressive spastic tetraparesis, visual impairments, bulbar dysfunction, and extrapyramidal signs. INAD patients usually die before their 10th birthday. In contrast to the severe defects in INAD, aNAD, and PARK14 patients have a later onset of symptoms, including progressive dystonia and parkinsonism. Cerebellar atrophy is a characteristic symptom in both INAD and aNAD (*Gregory et al., 2008*). The formation of spheroid structures in the nervous system is a typical neuropathological hallmark of PLAN (*Hedley-Whyte et al., 1968*). These spheroid structures contain numerous membranes as well as α-synuclein and ubiquitin (*Riku et al., 2013*) and are named tubulovesicular structures (TVSs) (*Sumi-Akamaru et al., 2015*). Moreover, Lewy bodies and phosphorylated tau-positive neurofibrillary tangles have also been observed in the nervous system of the PLAN patients (*Paisán-Ruiz et al., 2012*; *Riku et al., 2013*).

Mice that lack *Pla2g6* (genotype: *Pla2g6^{KO/KO}*) exhibit a normal lifespan but show a slowly progressive motor defect, first observed at around 1 year of age (*Malik et al., 2008*; *Shinzawa et al., 2008*). These mice show some phenotypes observed in INAD patients, including a progressive axonal degeneration (*Shinzawa et al., 2008*), cerebellar atrophy (*Zhao et al., 2011*), and the presence of TVS and α-synuclein-containing spheroids in the nervous system (*Malik et al., 2008*; *Shinzawa et al., 2008*). Transmission electron microscopy (TEM) revealed a swelling of the presynaptic membrane, synaptic degeneration, and mitochondrial inner membrane defects in these mice (*Beck et al., 2011*).

Another mouse model of INAD that harbors a spontaneous G373R missense mutation in *Pla2g6* (genotype: *Pla2g6^{G373R/G373R}*) was also identified. Homozygous *Pla2g6^{G373R/G373R}* mice express Pla2g6-G373R protein at a comparable level to wild-type littermates. The inheritance of this mutation is recessive and *Pla2g6^{G373R/G373R}* was proposed to be a severe loss of function allele (*Wada et al., 2009*). Interestingly, these homozygous *Pla2g6^{G373R/G373R}* mice only live for about 100 days and show severe behavioral and neuropathological phenotypes at a much earlier age than the mice that lack *Pla2g6* (*Wada et al., 2009*). In addition, no iron accumulation has been documented in the brain of both INAD mouse models.

We previously found that flies that lack *iPLA2-VIA* (the fly homolog of *PLA2G6*), abbreviated INAD flies, display slow progressive neurodegeneration, including severe bang-sensitivity, motor impairments as well as defects in vision (*Lin et al., 2018*). TEM revealed that INAD flies exhibit swelling of synaptic terminals, the presence of TVSs, the disruption of the mitochondrial inner membranes, and an obvious accumulation of lysosomes. The expression of human *PLA2G6* cDNA fully rescues these defects in flies. These data not only show functional conservation of *PLA2G6* throughout evolution but also suggest that introducing human *PLA2G6* into INAD mice or INAD/PARK14 patients may alleviate the progression of the disease.

We previously discovered that iPLA2-VIA binds to the retromer subunits, Vps26 and Vps35 (*Lin et al., 2018*). Loss of *iPLA2-VIA* reduces the level of Vps26 and Vps35, and impairs retromer function . This leads to an increase in trafficking to the lysosomes followed by an expansion of lysosomes in size and number. This in turn causes an elevation of ceramide levels, increasing membrane stiffness which may further impair retromer function, leading to a negative feed-forward amplification of the defects. Pharmacological or genetic manipulations that either reduce ceramide levels or activate the retromer robustly suppress the loss of *iPLA2-VIA*-associated neurodegenerative phenotypes in flies (*Lin et al., 2018*). Based on our previous findings, we proposed that impaired retromer-endolysosomal results in an increase in ceramide, which is toxic (*Lin et al., 2018*; *Lin et al., 2019*). Moreover, an enzymatic dead iPLA2-VIA can fully rescue the loss of *iPLA2-VIA*-associated defects highlighting the important role of iPLA2-VIA in regulating the endolysosomal pathway via its interaction with the retromer proteins.

The previous data raise several questions. Are the defective pathways we observed in flies similarly affected in INAD patient-derived cells as well as INAD mouse models? Can drugs that target ceramide metabolism and the endolysosomal pathway alleviate neurodegenerative phenotypes in INAD models? Finally, can the introduction of human *PLA2G6* cDNA rescue neurodegenerative phenotypes in INAD patient-derived cells and INAD mouse models? Here, we report a reduction of Vps35 and an elevation of ceramides and impaired endolysosomal trafficking and mitochondrial morphology in

INAD patient-derived cells as well as in the INAD mice, suggesting that these pathways are affected in the three species tested so far. We also assess 20 drugs in INAD flies and patient-derived cells and identify 4 drugs that improve lysosomal function and reduce elevated ceramides and suppress neurodegenerative phenotypes suggesting a causal relationship. Finally, we report the development of a gene therapy approach that alleviates neurodegenerative phenotypes and prolongs lifespan in the INAD mice providing potential therapeutic strategies to treat INAD and PARK14.

## Results

### *PLA2G6* is highly expressed in human iPSCs and NPCs, but not in skin fibroblasts

INAD patient-derived skin fibroblasts have been used in several studies to explore the molecular mechanisms of INAD (*Davids et al., 2016*; *Kinghorn et al., 2015*; *Sun et al., 2021*; *Villalón-García et al., 2022*). We tested the specificity of three commercially available PLA2G6 antibodies and found one antibody that recognizes PAL2G6 (sc-376563 Santa Cruz Biotechnology; *Figure 1—figure supplement 1A-B*). This antibody recognizes a band of the proper molecular weight in HEK-293T cells as well as induced pluripotent stem cells (iPSCs) and neural progenitor cells (NPCs) derived from a healthy person (*Figure 1—figure supplement 1A* and *Figure 1A*). Moreover, this band was absent in HEK-293T cells that express *PLA2G6* shRNAs and iPSCs and NPCs derived from an INAD patient with a PLA2G6-R70X variant (29-3; see below) (*Figure 1—figure supplement 1B* and *Figure 1A*). These data show that this antibody can specifically recognize endogenous PLA2G6.

To explore the expression levels of PLA2G6 in skin fibroblasts, we used skin fibroblasts derived from a healthy person as control. We observed very low expression levels of PLA2G6 (*Figure 1—figure supplement 1C*). We then obtained skin fibroblasts from two INAD patients and their parents, labeled Families 1 and 2 (*Figure 1—figure supplement 1—source data 1*), and observed that all six skin fibroblast lines, including four from unaffected parents and two from INAD patients, express no or very low levels of PLA2G6 (*Figure 1—figure supplement 1D*).

We previously showed that knocking down *PLA2G6* in Neuro-2A cells leads to the expansion of lysosomes. Moreover, loss of *iPLA2-VIA* in flies leads to the disruption of mitochondrial morphology (*Lin et al., 2018*). To assess the phenotypes associated with the patient-derived skin fibroblasts, we determined the levels of LAMP2, a lysosomal marker, as well as the morphology of mitochondria. The patient-derived skin fibroblasts (88101) from Family 1 show an elevation of LAMP2 levels and a disrupted mitochondrial morphology (*Figure 1—figure supplement 1D–E*). However, the patient-derived skin fibroblasts (1914560) from Family 2 do not show any of the phenotypes mentioned above (*Figure 1—figure supplement 1D–E*). Hence, the phenotypes are inconsistent in patient-derived skin fibroblasts. In summary, skin fibroblasts express no or very low levels of PLA2G6 and exhibit highly variable phenotypes making it difficult to interpret data derived from skin fibroblasts. We, therefore, opted to use iPSCs and neurons derived from these cells to characterize phenotypes associated with INAD.

### Ceramide accumulation, lysosomal expansion, and mitochondrial defects in INAD patient-derived NPCs and DA neurons

To assess the phenotypes associated with INAD patient-derived cells, we reprogramed lymphoblasts from a patient in Family 2 (*PLA2G6-R70X*) into iPSCs. We obtained two clones, 29-1 and 29-3. As mentioned above, 29-3 was used in *Figure 1A* to determine the expression of PLA2G6. To generate a control for the INAD patient-derived iPSCs, we used CRISPR technology to correct the variant in patient-derived iPSCs (29-1) to generate 29-2 (*Figure 1—figure supplements 2 and 3*). Hence, 29-1 and 29-2 are an isogenic pair of INAD patient-derived iPSCs. Patient and isogenic gene-corrected control lines were differentiated into ventral midbrain dopaminergic (DA) neurons. Each line uniformly expressed the positional transcription factor signature markers for ventral midbrain floor plate region neural progenitor cells (NPC) (OTX2+, LMX1A+, FOXA2+, NESTIN+, and are negative for the forebrain marker FOXG1; *Figure 1—figure supplement 2B–C*; *Nolbrant et al., 2017*). Upon further differentiation, these NPCs gave rise to a robust population of ventral midbrain DA neurons (TH+) with typical neuronal morphologies (*Figure 1—figure supplement 2D*). We then used the isogenic pair

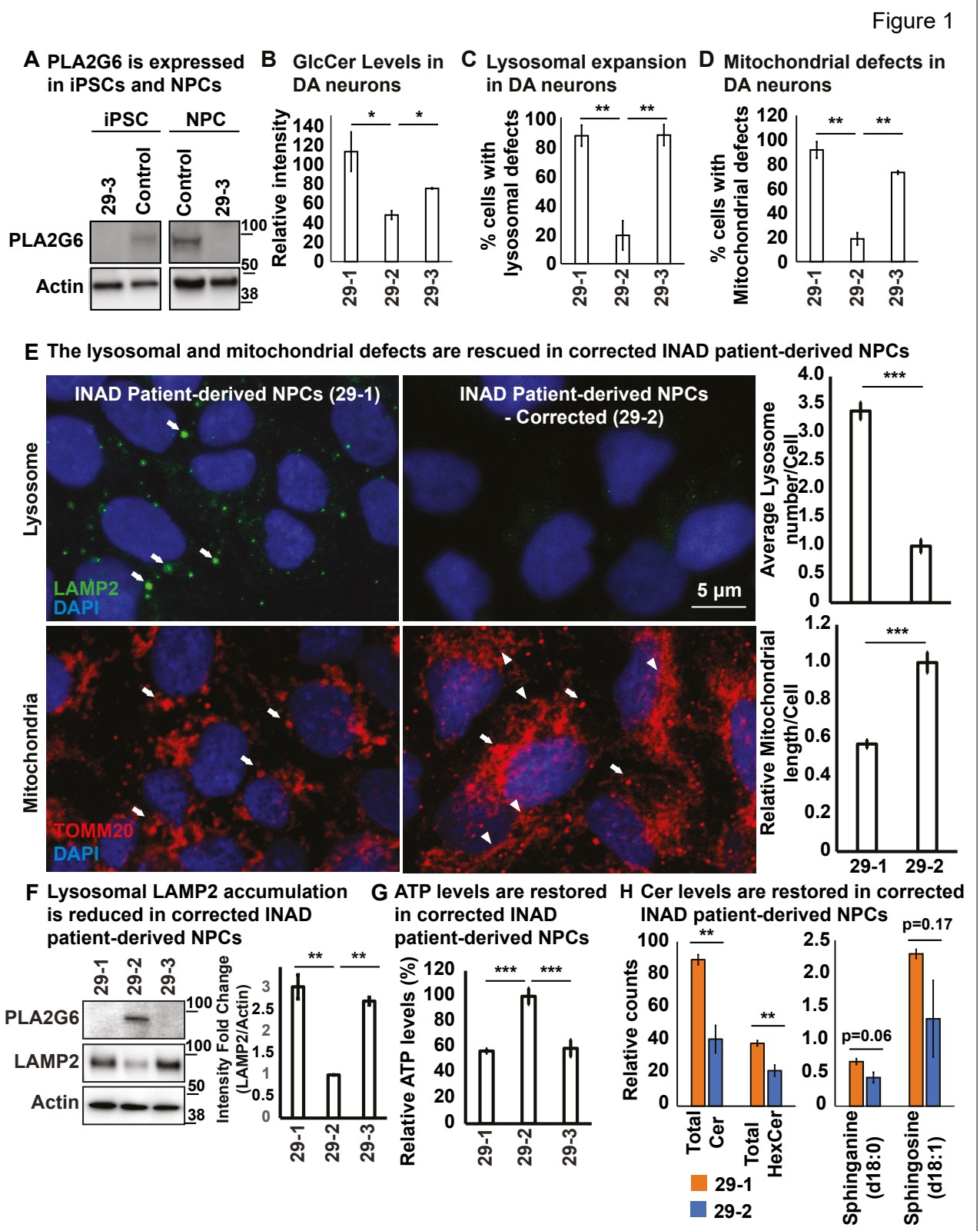

**Figure 1.** Ceramide accumulation, lysosomal expansion, and mitochondrial defects in INAD patient-derived NPCs and DA neurons. (**A**) PLA2G6 is expressed in iPSCs and NPCs using the sc-376563 antibody. The control iPSCs and NPCs were generated by reprograming a fibroblast line from a healthy person (GM23815; Coriell Institute). The 29-3 iPSCs and NPCs were generated from lymphoblasts from an INAD patient in Family 2 (*Figure 1—figure supplement 1D*). Actin was used as a loading control. (**B**) GlcCer levels in DA neurons (images in *Figure 1—figure supplement*

*Figure 1 continued on next page*

*Figure 1 continued*

*3C*) (n=8). (**C**) Lysosomal expansion in DA neurons (images in *Figure 1—figure supplement 3C*) (n=8–9). (**D**) Cells with mitochondrial defects (images in *Figure 1—figure supplement 3C*) (n=6–8). (**E**) The lysosomal and mitochondrial defects are rescued in edited INAD patient-derived NPCs. Immunofluorescence staining of the indicated INAD patient-derived NPCs. LAMP2 antibody (green; arrows) and TOMM20 antibody (red) were used to label lysosomes and mitochondria, respectively. DAPI (blue) labels cell nuclei. Scale bar=5 μm. The quantifications of the average lysosome number per cell (n=10) and the relative mitochondrial length per cell (n=10) are presented next to the images. (**F**) Lysosomal LAMP2 accumulation is reduced in edited INAD patient-derived NPCs. PLA2G6 antibody was used to detect the endogenous PLA2G6 in the indicated cellular lysates. LAMP2 antibody was used to assess lysosomal accumulation. Actin was used as a loading control. The intensity of the LAMP2/Actin is quantified at the right (n=3). (**G**) ATP levels are restored in edited INAD patient-derived NPCs. The relative amount of ATP are quantified in the indicated NPC lines (n=3). (**H**) Ceramide levels are reduced in the edited INAD patient-derived NPCs. The levels of the indicated ceramide and its derivatives are measured by lipidomic assays (n=3). Representative images are shown in this figure. Error bars represent SEM; *p<0.05; **p<0.01; ***p<0.001. INAD, infantile neuroaxonal dystrophy; NPC, neural progenitor cell.

The online version of this article includes the following source data and figure supplement(s) for figure 1:

**Source data 1.** PLA2G6 is expressed in iPSCs and NPCs.

**Source data 2.** The lysosomal defects are rescued in edited INAD patient-derived NPCs.

**Source data 3.** Lipidomic assay of INAD patient derived NPCs.

**Source data 4.** Raw gel images for *Figure 1*.

**Figure supplement 1.** Human skin fibroblasts express no or very low levels of PLA2G6 and exhibit highly variable phenotypes.

**Figure supplement 1—source data 1.** INAD patient skin fibroblast lines used in this study.

**Figure supplement 1—source data 2.** Raw gel images for *Figure 1—figure supplement 1*.

**Figure supplement 1—source data 3.** Skin fibroblasts derived from INAD patients exhibit highly variable phenotypes.

**Figure supplement 2.** Generation and quality control of INAD patient-derived NPCs and DA neurons.

**Figure supplement 3.** Ceramide accumulation, lysosomal expansion, and mitochondrial defects in INAD patient-derived DA neurons.

(29-1 and 29-2) of INAD patient-derived ventral midbrain NPCs as well as DA neurons differentiated from these NPCs in subsequent experiments.

Given that variants in *PLA2G6* also cause parkinsonism, we explored if there are defects associated with INAD patient-derived DA neurons. As shown in *Figure 1—figure supplement 3B*, we did not observe obvious overall cell morphological changes in INAD patient-derived (29-1 and 29-3) and genetically corrected DA neurons (29-2). We measured the levels of Glucosylceramide (GlcCer), and examined the morphology of lysosomes (LAMP2) and mitochondria (ATP5a). GlcCer is easily observed in both control- and patient-derived DA neurons (arrows in *Figure 1—figure supplement 3C; a-c*) but not in the undifferentiated NPCs (arrowheads in *Figure 1—figure supplement 3C; a-c*). This suggests that DA neurons, but not NPCs generate high levels of GlcCer. However, the GlcCer levels are two to three fold higher in INAD patient-derived DA neurons (29-1 and 29-3) than in corrected cells (29-2) (*Figure 1B* and *Figure 1—figure supplement 3C; a-c*). Furthermore, lysosomes are increased in number and enlarged in size in cells that carry the variant compared to corrected cells (*Figure 1C* and arrows in *Figure 1—figure supplement 3C; d-f*). Finally, we also observed enlarged mitochondria in the INAD patient-derived DA neurons (29-1 and 29-3) when compared to corrected cells (29-2) (*Figure 1D* and *Figure 1—figure supplement 3C; g-l*).

To extend our observations, we also examined the changes of lysosomal and mitochondrial morphology in NPCs. As shown in *Figure 1E*, we observed an increase in size and number of lysosomes in the INAD patient-derived NPCs (*Figure 1E*; arrows in upper panel; quantifications on the right). Moreover, the mitochondria in the corrected NPCs form a connected network (*Figure 1E*; arrowheads in lower panel). However, the mitochondria in the INAD patient-derived NPCs are enlarged and fragmented (*Figure 1E*; arrows in lower panel; quantifications on the right). To assess the levels of PLA2G6 and LAMP2, we performed western blots on NPCs (*Figure 1F*; quantified in the right panels). Both clones of the INAD patient-derived NPCs, 29-1 and 29-3, do not express PLA2G6 (*Figure 1F*; lanes 1 and 3, respectively). However, the genetically corrected cells obviously express PLA2G6 (*Figure 1F*; lane 2). Interestingly, LAMP2 levels are significantly upregulated in INAD patient-derived NPCs compared to corrected cells (*Figure 1F*), consistent with a lysosomal expansion. To assess the function of the mitochondria, we measured ATP levels in 29-1, 29-2, and 29-3 patient-derived NPCs. We found that the ATP levels are significantly higher in the genetically corrected cells

(29-2) than in the other two lines, indicating defective mitochondrial function in the patient-derived NPCs (*Figure 1G*).

Next, we performed lipidomic assays to measure the levels of Ceramides and their derivatives in the INAD patient-derived NPCs (29-1) and the genetically corrected cells (29-2). As shown in *Figure 1H*, we found a significant increase in total Cer and HexCer levels and they are about two fold higher in the patient-derived NPCs (*Figure 1H*; left panel and *Figure 1—source data 3*). However, although the levels of Sphinganine and Sphingosine are increased in the patient-derived NPCs, they are not statistically significant (*Figure 1H*; right panel and *Figure 1—source data 3*). The data show that INAD patient-derived NPCs exhibit an elevation of the HexCer (including GlcCer) and many different other types of ceramides. In summary, loss of *PLA2G6* leads to an elevation of GlcCer, expansion of lysosomes, and defects in mitochondrial dynamics and functions. Moreover, these phenotypes are reversed in genetically corrected cells, indicating a causative relationship.

## Ceramide accumulation, lysosomal expansion, and mitochondrial defects in mouse models of INAD

To study the neuropathology in mice, we obtained two INAD mouse models: mice that lack *Pla2g6* (*Pla2g6^{KO/KO}*) (*Bao et al., 2004*) and mice that carry a homozygous *G373R* point mutation (*Pla2g6^{G373R/G373R}*) (*Wada et al., 2009*). These two models exhibit similar neuropathological defects, as explained in the introduction, but the severity of the phenotypes is much stronger in the homozygous *Pla2g6^{G373R/G373R}* mice. The *Pla2g6^{KO/KO}* mice show a very slow progressive neurodegeneration, whereas the homozygous *Pla2g6^{G373R/G373R}* mice display an aggressive and quick neurodegenerative phenotype, yet they were both generated in a *C57BL/6* background. Note that three other *Pla2g6^{KO/KO}* have been generated in *C57BL/6* background and they all exhibit very similar very slow progressive phenotypes (*Malik et al., 2008*; *Shinzawa et al., 2008*; *Zhao et al., 2011*). Their lifespan is normal and the first rotarod assay defects are observed at ~300 days. Moreover, three other models have been described: *Pla2g6^{D331Y/D331Y}*, *Pla2g6^{R748W/R748W}* as well as a 5′ UTR transposon insertion *Pla2g6^{IAP/IAP}* (*Chiu et al., 2019*; *Strokin et al., 2012*; *Sun et al., 2021*). Both point mutation lines, *Pla2g6^{D331Y/D331Y}* and *Pla2g6^{R748W/R748W}*, express mutant Pla2g6 at a level comparable to their wild-type littermates (*Chiu et al., 2019*; *Sun et al., 2021*). These latter three different mouse models cause very similar and severe phenotypes when compared to the null alleles. The *Pla2g6^{IAP/IAP}* mice produce ~10% of the wild-type protein (*Strokin et al., 2012*). Given that four mouse models that lack the protein have much less severe phenotypes than the three models that produce Pla2g6 protein, we propose that a compensatory pathway is activated upon a complete loss of *Pla2g6* gene/protein during development. Hence, to minimize genetic background issues and avoid the possible activation of a compensatory pathway, we generated and characterized transheterozygous animals (*Pla2g6^{KO/G373R}*).

To characterize the *Pla2g6^{KO/G373R}* mice, we performed rotarod and lifespan assays. As shown in *Figure 2A*, the control littermates (*Pla2g6^{+/+}*—Wild-type; *Pla2g6^{KO/+}*; and *Pla2g6^{+/G373R}*) show similar normal performances on rotarod assays over a period of 160 days. Homozygous *Pla2g6^{G373R/G373R}* mice show the first signs of rotarod defects between 60 and 70 days of age (*Figure 2A*) and die within 100 days (*Figure 2B*). The *Pla2g6^{KO/G373R}* mice show the first signs of rotarod defects between 80 and 90 days of age (*Figure 2A*) and die at ~150 days of age (*Figure 2B*). Hence, we decided to mainly focus on the *Pla2g6^{KO/G373R}* mice in the following experiments but used the *Pla2g6^{G373R/G373R}* mice as a comparison in some assays.

We previously showed that loss of fly homolog of *PLA2G6* leads to an elevation of ceramides including GlcCer (*Lin et al., 2018*). GlcCer accumulation is also observed in flies that lack the fly homolog of *GBA1* (*Wang et al., 2022a*), the gene that causes Gaucher disease (*Sidransky, 2004*; *Wong et al., 2004*). We performed immunostaining in the cerebellum and midbrain of the *Pla2g6^{KO/G373R}* and homozygous *Pla2g6^{G373R/G373R}* animals. As shown in *Figure 2C*, the GlcCer levels are highly upregulated in the Purkinje cells (*Figure 2C*; upper panel) and mid-brain cells (*Figure 2C*; lower panel) of both INAD mouse models when compared to controls, again showing defective ceramide metabolism. Moreover, we performed lipidomic assays to measure the levels of total Cer, total HexCer as well as Sphinganine and Sphingosine in the cerebellum of the *Pla2g6^{KO/G373R}* mice. However, unlike in flies and human cells, we did not observe an obvious change in HexCer as well as other ceramides (*Figure 2—figure supplement 1D* and *Figure 2—source data 1*). Given that the GlcCer antibody shows obvious changes that are consistent among the three species, we argue that the lipidomic

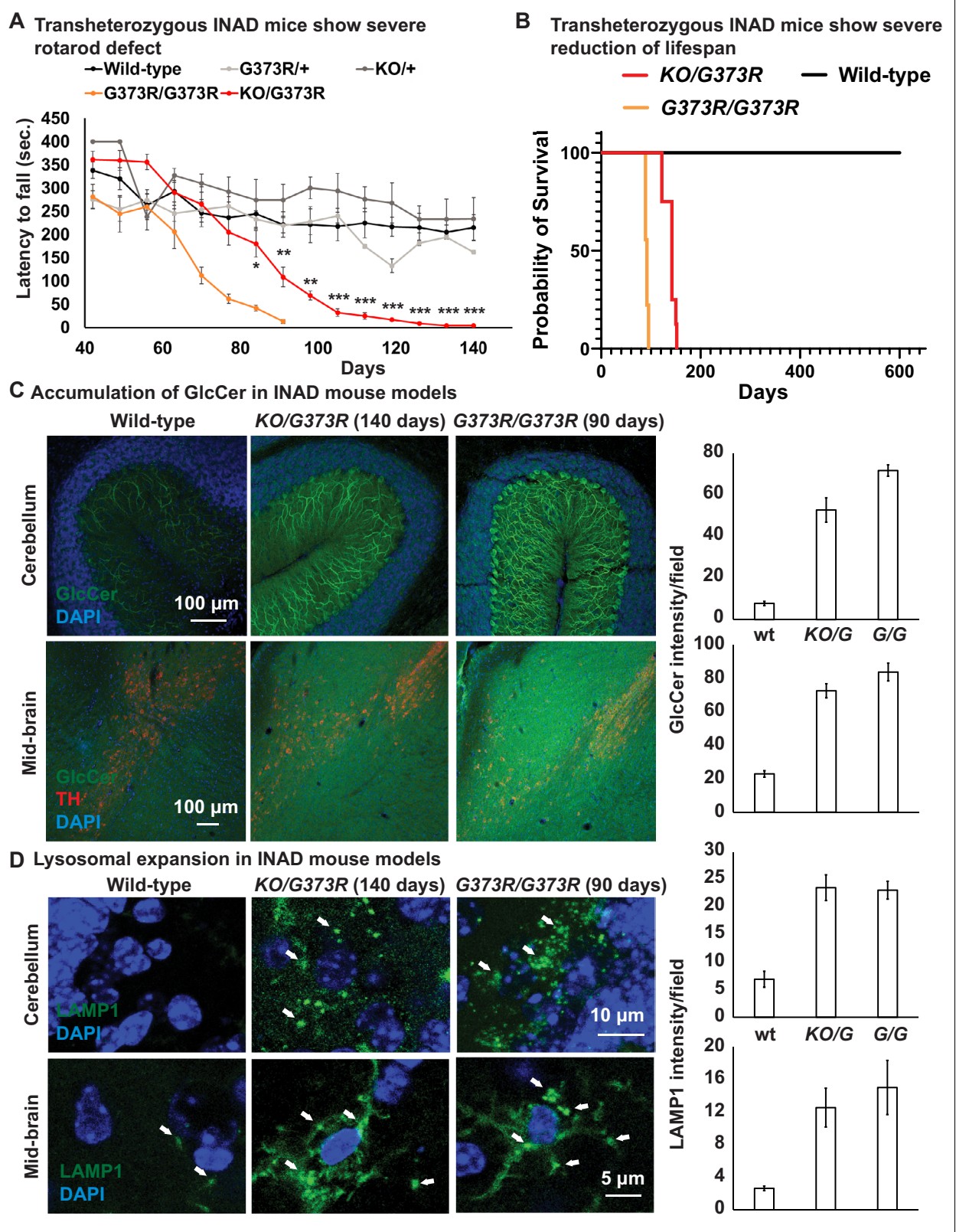

**Figure 2.** Ceramide accumulation, lysosomal expansion, and mitochondrial defects in mouse models of INAD. (**A**) Both *Pla2g6*<sup>G373R/G373R</sup> and *Pla2g6*<sup>KO/G373R</sup> mice show severe rotarod defect. Rotarod performance of mice with the indicated genotypes was measured weekly. Wild-type (n=6); *Pla2g6*<sup>+/G373R</sup> (n=10); *Pla2g6*<sup>G373R/G373R</sup> (n=7); *Pla2g6*<sup>KO/+</sup> (n=4); and *Pla2g6*<sup>KO/G373R</sup> (n=7). Error bars represent SEM. (**B**) *Pla2g6*<sup>G373R/G373R</sup> and *Pla2g6*<sup>KO/G373R</sup> mice fantile neuroaxonal dystrophy. A disease characterized by altered terminal axons and synaptic enshow a severe reduction of lifespan. Wild-type (n=6);

*Figure 2 continued on next page*

*Figure 2 continued*

*Pla2g6^{G373R/G373R}* (n=9); and *Pla2g6^{KO/G373R}* (n=8). (**C**) Accumulation of GlcCer in *PLA2G6^{KO/G373R}* and *PLA2G6^{G373R/G373R}* mice. Immunofluorescent staining of mouse cerebella and midbrain regions of the indicated genotypes. GlcCer antibody (green) was used to assess the levels of GlcCer. TH (Tyrosine Hydroxylase, red) antibody labels DA neurons in the midbrain region. Scale bar=100 μm. All assays were conducted in blind of genotypes and treatments. (**D**) Lysosomal expansion in INAD in *Pla2g6^{KO/G373R}* and *Pla2g6^{G373R/G373R}* mice. Immunofluorescent staining of mouse cerebella and midbrain regions of the indicated genotypes. LAMP1 antibody (green; arrows) labels lysosomes. DAPI (blue) labels nuclei. Representative images are shown in this figure. Quantifications are next to the images (n=3). Scale bar=10 μm (cerebellum) or 5 μm (midbrain). DA, dopaminergic; INAD, infantile neuroaxonal dystrophy.

The online version of this article includes the following source data and figure supplement(s) for figure 2:

**Source data 1.** Lipidomic assay of the cerebellum of INAD mice.

**Figure supplement 1.** *Pla2g6^{G373R/G373R}* mice show disrupted mitochondria, and increased MVB and TVS in Purkinje neurons.

assays are not sensitive enough to detect the increase in GlcCer possibly because only Purkinje cells seem to be affected.

We next explored the morphology of lysosomes in the cerebellum and midbrain of the *Pla2g6^{KO/G373R}* and homozygous *Pla2g6^{G373R/G373R}* animals. As shown in *Figure 2D*, we observed an expansion of LAMP1, a lysosomal marker, in the Purkinje cells in the cerebellum (*Figure 2D*; upper panel) and neurons of the midbrain region (*Figure 2D*; lower panel). Hence, lysosomal expansion is a common feature of all models of INAD.

We next performed TEM to assess the morphology of the mitochondria in the cerebellum of the homozygous *Pla2g6^{G373R/G373R}* animals. As shown in *Figure 2—figure supplement 1A*, the morphology of mitochondria is disrupted in Purkinje cells of the homozygous *Pla2g6^{G373R/G373R}* animals (*Figure 2—figure supplement 1A–C*). Moreover, we also observed a very significant increase in multivesicular bodies when compared to control animals (*Figure 2—figure supplement 1B–C*), consistent with an endolysosomal defect. Finally, these mice also exhibit TVSs (*Figure 2—figure supplement 1B–C*), a hallmark of *Pla2g6* mutant animals (*Sumi-Akamaru et al., 2015*).

In summary, *Pla2g6* mutant mice exhibit a ceramide accumulation, lysosomal enlargement as well as mitochondrial defects. These data are consistent with the fly model of INAD (*Lin et al., 2018*) and patient-derived NPCs (*Figure 1E–H*) and DA neurons (*Figure 1—figure supplement 3C*). Hence, we argue that these defects are evolutionary conserved and may play a critical role in the pathogenesis of INAD/PARK14.

## Ambroxol, Azoramide, Desipramine, or Genistein alleviate neurodegenerative phenotypes in INAD flies and patient-derived NPCs

We next tried to identify therapeutic strategies for INAD. We used INAD flies in a primary screen and then tested the drugs that improve the phenotypes in flies in INAD patient-derived cells. Based on a review of the literature, we identified 20 drugs that have been reported to control or affect sphingolipid metabolism, endolysosomal trafficking and drugs that are being tested to treat Parkinson's disease (*Figure 3—source data 2*; see references in Table 4). We previously reported that INAD flies show severe bang-sensitivity at 15 days of age (*Lin et al., 2018*). This assay is a quick and sensitive assay that measures the propensity of flies to seize upon shock (*Figure 3A*). Wild-type flies right themselves in less than a second after a vortex paradigm. In contrast, bang-sensitive flies are paralyzed for an extended period of time before they can right themselves. We used this assay as our primary screen assay. As shown in *Figure 3A*, wild-type flies are not bang-sensitive. In contrast, INAD flies show severe bang-sensitivity and require ~20 s to recover (*Figure 3A*). We identified drugs that worsen the bang-sensitivity, including Fingolimod, Ozanimod, Fumonisin, Hydrochloroquine, Lonafarnib, Omigapil, and Taurine (*Figure 3A*). Another set of drugs did not affect bang-sensitivity, including Miglustat, Ibiglustat, NCGC607, Rapamycin, VER-155008, Deoxygalactonojirimycin, CuATSM, and Metformin (*Figure 3A*). However, five drugs suppressed bang-sensitivity, including Ambroxol, Genistein, PADK, ML-SA1, and Azoramide (*Figure 3A*). To confirm the rescue effect, we retested the drugs that suppressed the bang-sensitivity at a dosage that is 10 times higher than the dosage used in the first round. Each drug exhibited a dose-dependent increase in rescue activity, suggesting that the ability to suppress bang-sensitivity is dose-dependent (*Figure 3A*). To assess if these drugs modify the phenotypes in INAD patient-derived NPCs we probed Western blots of the treated cells for LAMP2. As shown in *Figure 3B and C*, patient-derived NPCs (29-1) do not express PLA2G6 and exhibit elevated levels of LAMP2.

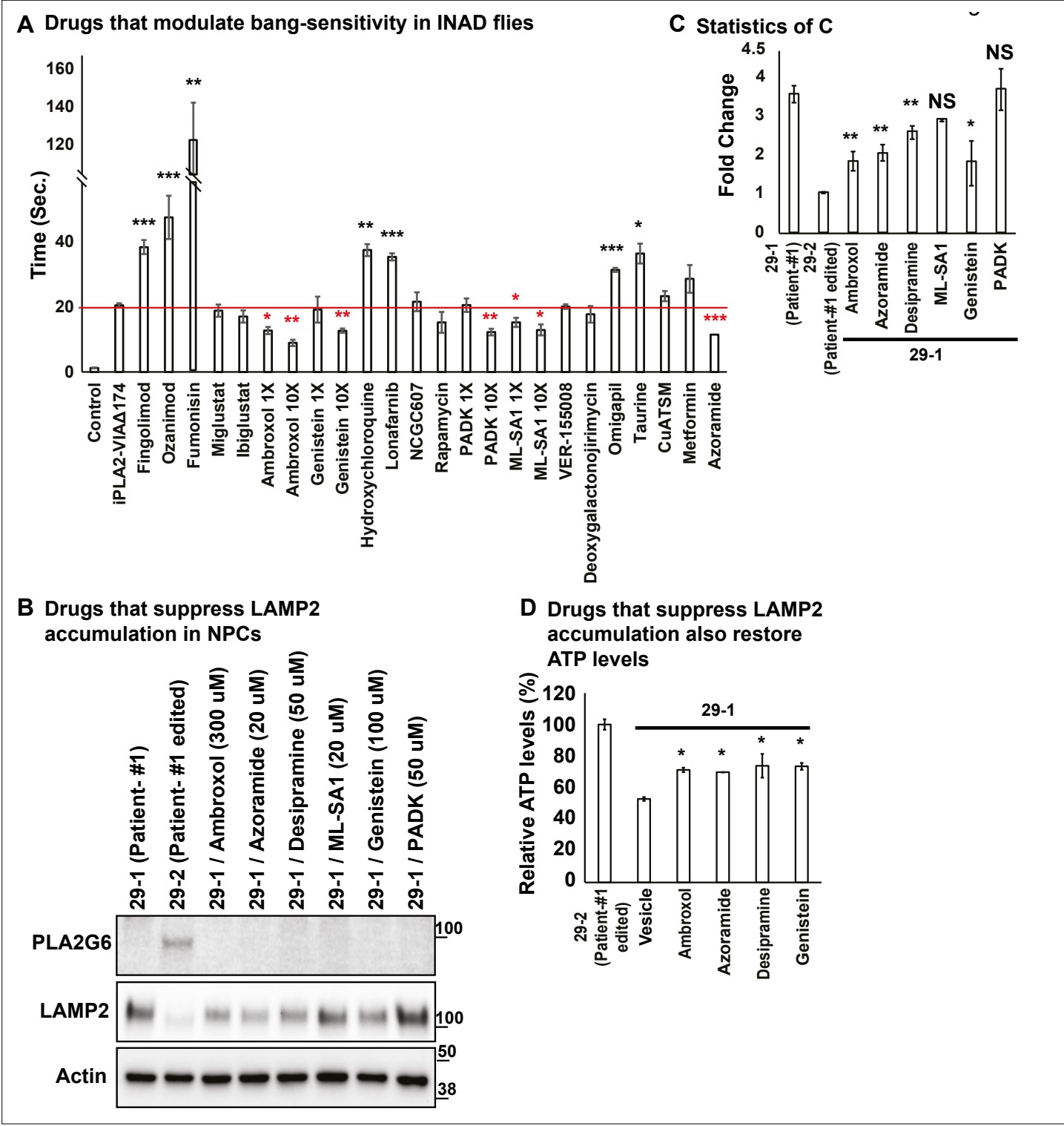

**Figure 3.** Ambroxol, Azoramide, Desipramine, and Genistein alleviate neurodegenerative phenotypes in INAD flies and patient-derived NPCs. (**A**) Bang-sensitivity was used as a primary readout to select drugs that suppress neurodegeneration. Bang-sensitivity of control or INAD flies fed with the indicated drugs. Error bars represent SEM (n=3; 20 flies per assay). Redline highlights the time required for INAD flies to recover from bang-induced paralysis. Red '*' indicates drugs that significantly suppress bang-sensitivity. Black '*' indicates drugs that significantly promote bang-sensitivity (the p values were calculated between *iPLA2-VIA^Δ174* and the indicated treatment). (B) Using INAD patient-derived NPCs to select drugs that suppress LAMP2 accumulation. PLA2G6 antibody was used to detect the endogenous PLA2G6 in the indicated cellular lysates. The intensity of the LAMP2/Actin is quantified at (C) (n=3). (D) Drugs that suppress LAMP2 accumulation also restore ATP levels. The relative amounts of ATP are measured in the indicated

*Figure 3 continued on next page*

Figure 3 continued

NPC lines with or without the treatment of the selected drugs (n=3). Error bars represent SEM; *p<0.05; **p<0.01; ***p<0.001; NS, not significant (the p values were calculated between the untreated patient NPCs (29-1) and the indicated treatment). References in *Figure 3A*: (*Aflaki et al., 2016*; *Agostini et al., 2021*; *Alfonso et al., 2005*; *Desai et al., 2002*; *Fu et al., 2015*; *Hernandez et al., 2019*; *Hung et al., 2012*; *Hwang et al., 2019*; *Ke et al., 2020*; *Khanna et al., 2010*; *Liu et al., 2008*; *Magalhaes et al., 2018*; *Mauthe et al., 2018*; *Mistry et al., 2018*; *Moskot et al., 2014*; *Olanow et al., 2006*; *Rosen and Liao, 2003*; *Scott et al., 2016*; *Shen et al., 2012*; *Wang et al., 2021*; *Yang and Tohda, 2018*; *Zhu et al., 2019*). INAD, infantile neuroaxonal dystrophy; NPC, neural progenitor cell.

The online version of this article includes the following source data and figure supplement(s) for figure 3:

Source data 1. Ambroxol, Azoramide, Desipramine, or Genistein reduce LAMP2 levels in patient derived NPCs.

Source data 2. Selected drugs tested in an INAD fly model.

Source data 3. Raw gel images for *Figure 3*.

Figure supplement 1. Ambroxol, Azoramide, Desipramine, or Genistein suppress ERG defects and the loss of photoreceptor in INAD flies.

Correcting the variant (29-2) strongly reduces LAMP2 levels. Importantly, Ambroxol, Azoraminde, and Genistein, significantly reduce LAMP2 levels in the patient-derived NPCs (*Figure 3B and C*).

We previously showed that Myriocin, R55, and Desipramine reduced ceramide levels as well as the lysosomal expansion, and alleviated bang-sensitivity in the INAD fly model (*Lin et al., 2018*). Given that Myriocin is toxic in vertebrates and that R55 poorly penetrates the blood-brain barrier we tested Desipramine, an FDA-approved drug, in the INAD patient-derived NPCs. As shown in *Figure 3B and C*, Desipramine also reduces LAMP2 levels in patient-derived NPCs. We further tested whether the drugs, Ambroxol, Azoraminde, Desipramine, or Genistein, restore mitochondrial functions. As shown in *Figure 3D*, the ATP levels were restored upon exposure to the drugs. Taken together, these data show that the drugs not only reduce lysosomal expansion but also restore mitochondrial function.

We have previously shown that INAD flies also exhibit morphological as well as functional defects of photoreceptors (*Lin et al., 2018*). We therfore tested whether Ambroxol, Azoraminde, Desipramine, or Genistein can rescue these defects. As shown in *Figure 3—figure supplement 1A–C*, the obvious defects in electroretinograms (ERGs) profiles, including the loss of light coincident receptor potentials (LCRPs) and on-transients, are significantly improved by the drugs. In addition, the loss of photoreceptors phenotypes is also reduced in INAD flies exposed to the drugs (*Figure 3—figure supplement 1D–E*). In summary, we identified four drugs that suppress the loss of *PLA2G6*-induced phenotypes in flies and INAD patient-derived cells: Ambroxol, Azoraminde, Desipramine, and Genistein.

## Expression of human *PLA2G6* restores lysosomal and mitochondrial morphology defects in INAD patient-derived NPC lines

We previously showed that whole-body expression of human *PLA2G6* in INAD flies fully rescued the neurodegenerative phenotypes and lifespan (*Lin et al., 2018*). In contrast, neuronal expression of human *PLA2G6* strongly suppressed neurodegenerative phenotypes, but did not prolong lifespan. These data suggest that even though *PLA2G6* plays an important role in the nervous system, it is also required in cells other than neurons. We, therefore, surmised that a gene therapy approach in mice should attempt the delivery of human *PLA2G6* into as many cell types as possible. We, therefore, designed an AAV-based gene therapy construct, *AAV-EF1a-PLA2G6* (*Figure 4A*). We used elongation factor EF-1 alpha (EF1a), a ubiquitous promoter, to express *PLA2G6* as broadly as possible. We also designed two constructs, *Lenti-CMV-PLA2G6*, and *AAV-EF1a-EGFP* as controls (*Figure 4A*). The lentivirus-based construct allowed quite elevated levels of expression and was used to test toxicity when *PLA2G6* is highly expressed in NPCs. The *AAV-EF1a-EGFP* construct was used to track viral transduction and expression efficiency. Wild-type mice injected with the *AAV-EF1a-EGFP* (serotype) construct via intracerebroventricular (ICV) and intravenous (IV) injections at P40 express EGFP in numerous tissues, including the cerebellum, olfactory bulb, cerebral cortex, mid-brain region, spinal cord, sciatic nerve, heart, and liver (*Figure 4—figure supplement 1*). However, EGFP is not expressed in the eyes or the uninjected animals (*Figure 4—figure supplement 1*). These data show that *AAV-EF1a-EGFP* is broadly expressed in many tissues when the virus is delivered via ICV and IV. However, not all cells express EGFP based on this assay.

We then conducted a pilot experiment in HEK-293T cells to assess the expression of the protein using these constructs. As shown in *Figure 4B*, HEK-293T expresses very low levels of endogenous

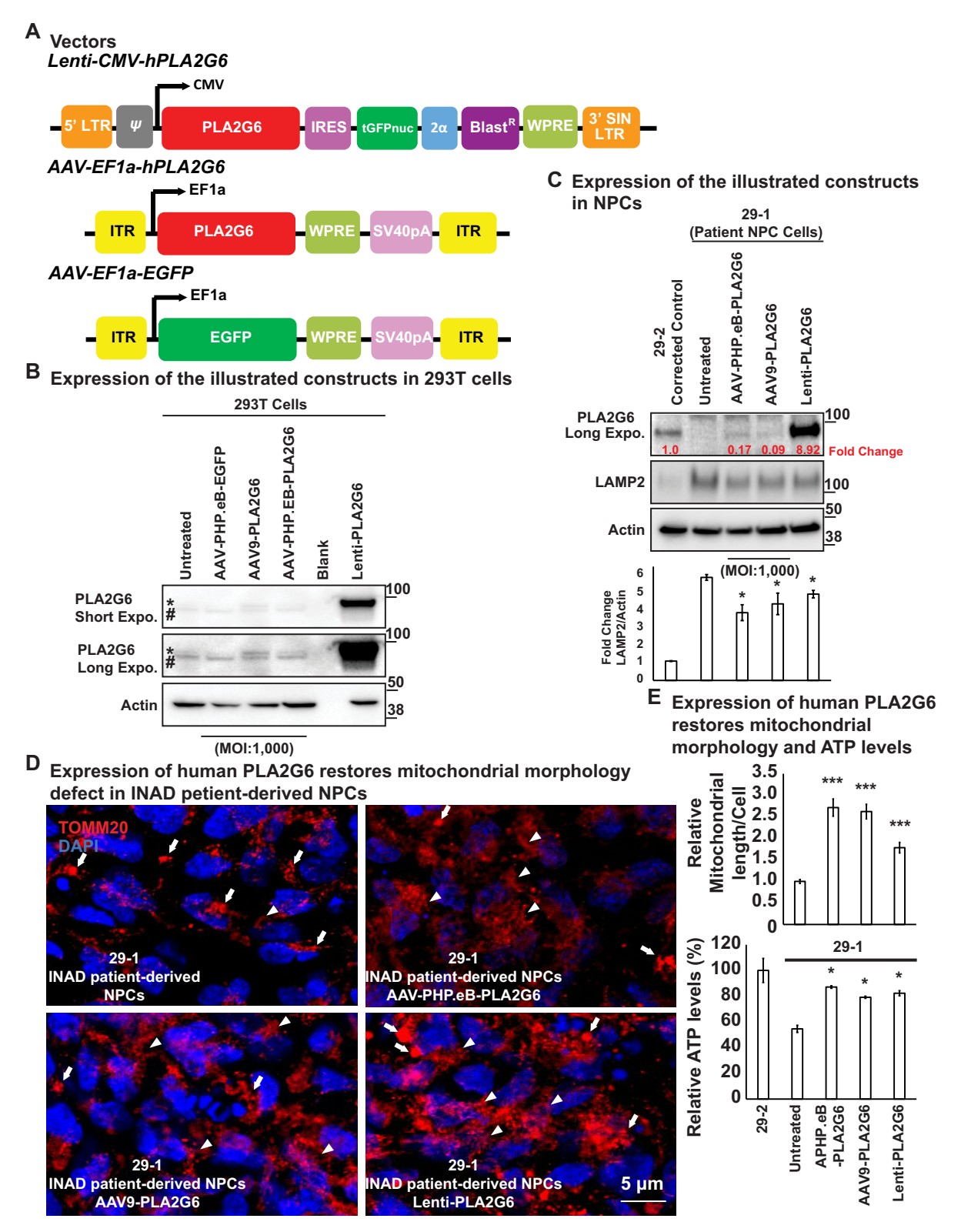

**Figure 4.** Expression of human PLA2G6 restores lysosomal and mitochondrial morphology defects in INAD patient-derived NPC lines. (**A**) Vectors/constructs. (**B**) Expression levels of the constructs (**A**) in 293T cells. PLA2G6 antibody was used to detect the endogenous PLA2G6 of cellular lysates. * represents endogenous PLA2G6. # indicates a nonspecific band. All AAV constructs were used to infect 293T cell using a MOI of 1000. The MOI of the Lenti-viral-based construct was not determined. (**C**) Expression levels of the illustrated constructs (**A**) in NPCs. AAV constructs were used to

*Figure 4 continued on next page*

*Figure 4 continued*

infect NPCs at a MOI of 1000. The MOI of the Lenti-viral-based construct was not determined. The intensity of LAMP2/Actin is quantified below (n=3). (**D**) Expression of human PLA2G6 restores mitochondrial morphology defects in INAD patient-derived NPCs. Arrows indicate the fragmented and enlarged mitochondria. Arrowheads indicate the normal elongated network of mitochondria. (**E**) Expression of human PLA2G6 restores mitochondrial morphology and ATP levels. The length of the mitochondria in (**D**) is quantified in (**E**) (n=10; upper panel). The ATP levels are measure in the lower panel (n=3). Representative images are shown in this figure. Error bars represent SEM; *p<0.05; ***p<0.001 (the p values were calculated between the untreated patient NPCs (29-1) and the indicated treatment). INAD, infantile neuroaxonal dystrophy; MOI, multiplicity of infection; NPC, neural progenitor cell.

The online version of this article includes the following source data and figure supplement(s) for figure 4:

**Source data 1.** Expression of human PLA2G6 in INAD patient derived NPCs.

**Source data 2.** Expression of human PLA2G6 restores lysosomal defects in INAD patient-derived NPCs.

**Source data 3.** Raw gel images for *Figure 4*.

**Figure supplement 1.** Expression of EGFP in the indicated sites/tissues in *AAV-EF1a-EGFP* injected (ICV and IV at P40) (Left) or uninjected (Right) wild-type mice.

PLA2G6 (upper bands marked by a *). Note that we also observed a nonspecific band (lower bands marked by #). As expected, *Lenti- CMV-PLA2G6* induces very high levels of expression of human PLA2G6 in HEK-293T cells (stars in *Figure 4B*; last lane). However, expression of human PLA2G6 by infecting cells with *AAV-EF1a-PLA2G6* packaged in either the AAV-PHP.eB or AAV9 serotype is very low, even when we used a very high multiplicity of infection (MOI of 1000) (stars in *Figure 4B*; lanes 3 and 4).

Upon showing that these constructs can be expressed in HEK-293T cells, we tested their expression and function in the patient-derived NPCs (*Figure 4C*). Expression of PLA2G6 is easily detectable in NPC cells. Moreover, the Lenti-CMV-PLA2G6 derived protein is expressed at very high levels (~9-fold higher than the endogenous levels) in the NPCs (*Figure 4C*; last lane). However, expression of human PLA2G6 driven by *AAV-EF1a-PLA2G6* construct packaged in AAV-PHP.eB or AAV9 serotype is very low (~10% of the endogenous levels) (*Figure 4C*; lanes 3 and 4). Interestingly, the LAMP2 levels are reduced by about 10–20% (*Figure 4C*; bottom) in all three conditions, suggesting that even very low levels may have a beneficial effect.

We also examined mitochondrial morphology in these cells. As shown in *Figure 4D*, the patient-derived NPCs show enlarged and fragmented mitochondrial morphology (*Figure 4D*; upper left; arrows). Delivery of AAV-PHP.eB-PLA2G6 (*Figure 4D*; upper right; arrowheads) or AAV9-PLA2G6 (*Figure 4D*; lower left; arrowheads) strongly suppress this morphological abnormality as we observe many elongated mitochondria. In contrast, the Lenti-CMV-PLA2G6, which expresses very high levels of human PLA2G6, only partially rescues mitochondrial morphological abnormalities (*Figure 4D*; lower right; arrowheads). This suggests that very high levels of expression of PLA2G6 may be somewhat toxic. We also measured the ATP levels in the NPCs and found that the ATP levels are restored in the treated NPCs, arguing that mitochondrial function is at least partially restored (*Figure 4E*; lower panel).

In summary, both AAV-PHP.eB-PLA2G6 and AAV9-PLA2G6 induce low levels of expression of human PLA2G6 in NPCs (10–20% of the endogenous levels), and can partially alleviate lysosomal expansion as well as mitochondrial defects.

## Expression of human *PLA2G6* restores Vps35 levels in INAD patient-derived NPCs

We previously showed that PLA2G6 interacts with retromer proteins, Vps26 and Vps35. This interaction is important to maintain the proper levels of Vps26 and Vps35 (*Lin et al., 2018*). Loss of *PLA2G6* reduces the levels of Vps26 and Vps35 and impairs retromer function as well as the recycling of proteins and lipids. This leads to an accumulation of proteins and ceramides in the lysosomes and causes a lysosomal expansion. To assess whether the retromer is affected in the patient-derived cells, we assessed the levels of Vps35 in INAD patient-derived (29-1) and genetically corrected NPCs (29-2). We found that the number of Vps35 punctae is much higher in the genetically corrected NPCs (29-2) than in INAD patient-derived NPCs (29-1) (*Figure 5A, a–b and B*), consistent with our observations in flies that PLA2G6 increase the levels of Vps35 (*Lin et al., 2018*).

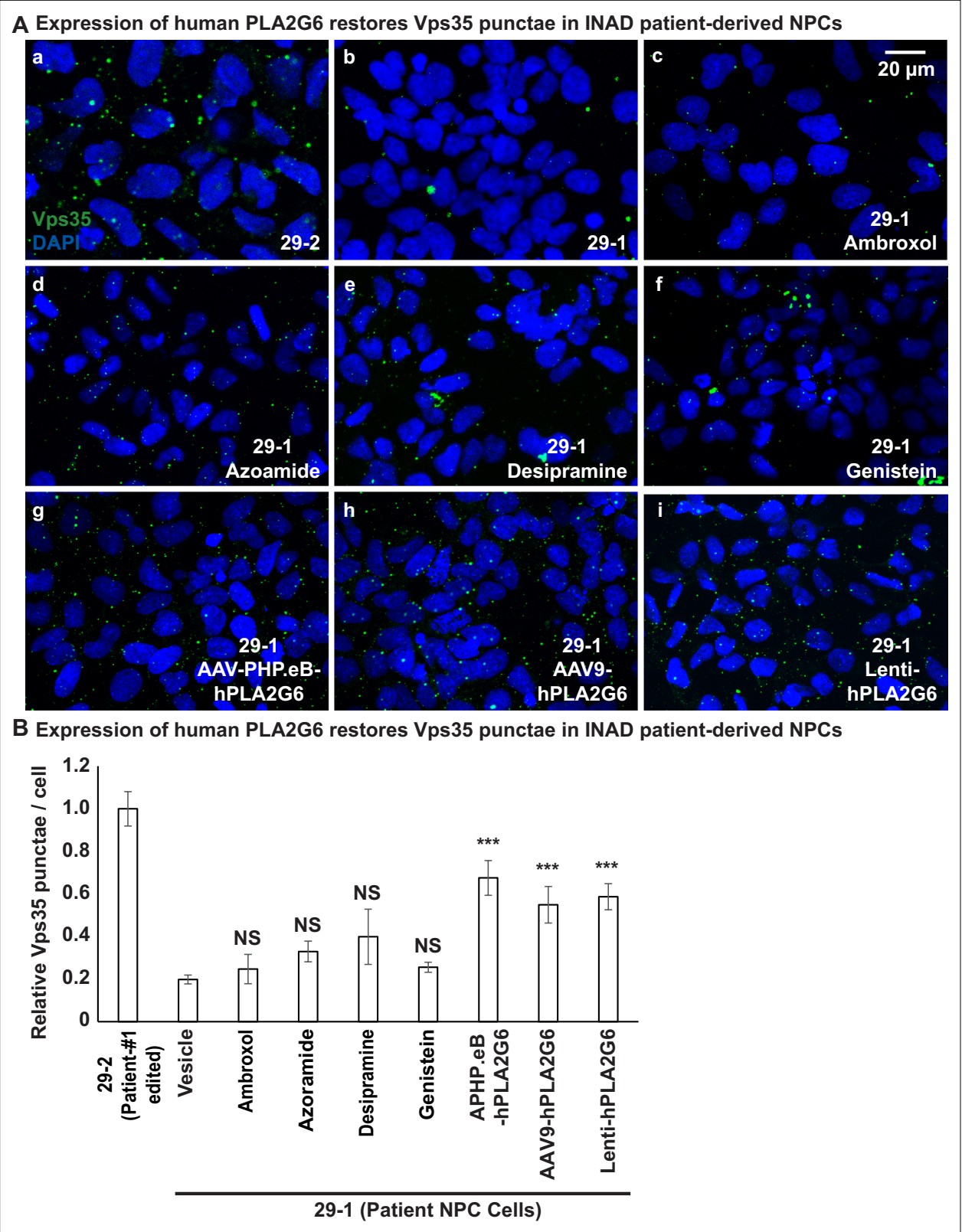

**Figure 5.** Expression of human PLA2G6 restores the number of Vps35 punctae in INAD patient-derived NPCs. **A** (**a, b**) The number of Vps35 punctae is higher in the genetically corrected NPCs (29-2). Immunofluorescent staining of NPCs with the indicated genotypes and treatments. Vps35 antibody (green; arrows) labels retromers. DAPI (blue) labels nuclei. (**c–f**) The number of Vps35 punctae is not affected in the patient-derived NPCs (29-1) treated with the selected drugs. (**g–i**) The expression of human PLA2G6 restores the number of Vps35 punctae in INAD patient-derived NPCs.

*Figure 5 continued on next page*

*Figure 5 continued*

(**B**) Quantifications of the Vps35 punctae in (**A**) (n=8). Representative images are shown in this figure. Scale bar=20 μm. Error bars represent SEM; ***p<0.001; NS, not significant (the p values were calculated between the untreated patient NPCs (29-1) and the indicated treatment). INAD, infantile neuroaxonal dystrophy; NPC, neural progenitor cell.

Next, we tested whether the selected drugs or the expression of human *PLA2G6* restores Vps35 levels in INAD patient-derived NPCs. Interestingly, the addition of Ambroxol, Azoraminde, Desipramine, or Genistein to the patient-derived NPCs (29-1) did not restore the Vps35 levels (*Figure 5A, a–f and B*), suggesting that the drugs function downstream of the retromer. Ambroxol promotes GCase activity to degrade GlcCer in lysosomes (*Magalhaes et al., 2018*), Azoraminde promotes protein folding (*Fu et al., 2015*), Desipramine blocks the ceramide salvage pathway (*Jenkins et al., 2011*), and Genistein enhances lysosome biogenesis (*Moskot et al., 2014*). These actions are downstream of retromer function. In contrast to the drugs, expression of the human *PLA2G6* restores Vps35 levels (*Figure 5A, a, b, g–i and B*), consistent with the model that PLA2G6 binds to Vps35 and Vps26 to maintain the levels of the retromer complex. In summary, expression of the human *PLA2G6*, but not the identified drugs restores the Vps35 levels.

## Pre-symptomatic injection of AAV-PHP.eB-PLA2G6 suppresses rotarod defect and prolongs lifespan in *Pla2g6^{KO/G373R}* INAD mice

To determine if delivery of human *PLA2G6* alleviates the phenotypes in INAD mouse model, we injected five *Pla2g6^{KO/G373R}* mice with AAV-PHP.eB-EF1a-PLA2G6 construct at postnatal day 40 via ICV (4.5×10^{10} GC) and IV (5×10^{11} GC) injections (*Figure 6A*; blue line). In an independent litter, we injected two more *Pla2g6^{KO/G373R}* mice at postnatal day 40 via ICV only (*Figure 6A*; purple line). Un-injected *Pla2g6^{KO/G373R}* mice displayed the first signs of rotarod impairment at ~90 days of age (*Figure 6A*; red line). In contrast, the ICV and IV injected *Pla2g6^{KO/G373R}* mice (*Figure 6A*; blue line) did not show signs of rotarod impairment until 130–140 days, a 50–60 day delay in the onset of the rotarod defects. However, even though the ICV-injected mice show some improvement in rotarod performance, statistical analyses did not show significance at most time points (*Figure 6A*; purple line). Note that injection of the AAV-PHP.eB-EF1a-PLA2G6 construct in wild-type littermates did not cause any obvious change in rotarod performance.

We also assessed body weight and lifespan. All injected *Pla2g6^{KO/G373R}* mice (both the ICV+IV injected group and IV injected group) exhibit a sudden body weight drop at ~190 days, 2 weeks before they die (*Figure 6B*). This is ~70 days later than the uninjected *Pla2g6^{KO/G373R}* animals, which show an obvious drop in body weight at ~120 days (*Figure 6B*). Five of the injected mice died by 210 days and two animals lived for over 300 days (*Figure 6C*; blue line). *Figure 6* We scarified these two animals to assess the molecular changes . Hence, the injected animals live an average of at least 65 days longer than the uninjected *Pla2g6^{KO/G373R}* mice (*Figure 6C*; red line). In summary, our data suggest that (1) expression of human PLA2G6 in mice is safe; (2) expression of human PLA2G6 in adult pre-symptomatic *Pla2g6^{KO/G373R}* mice delays the onset of defects; and (3) promoting broad expression of PLA2G6 using ICV and IV injections is more effective than ICV delivery only.

## Pre-symptomatic injection of AAV-PHP.eB-PLA2G6 reduces the levels of GlcCer and LAMP2, and increases the levels of Vps35 in *Pla2g6^{KO/G373R}* INAD mice

To explore the ability of AAV-PHP.eB-PLA2G6 injections to rescue the observed defects in the cerebella of the mice, we scarified two injected *Pla2g6^{KO/G373R}* mice at ~P300. These two mice were not able to stay on the rod in a rotarod assay but behaved relatively normal in their home cage (see *Video 1*). As shown in *Figure 7A*, we found that the elevated GlcCer levels are suppressed in the P300 Purkinje cells of these two animals when compared to the uninjected *Pla2g6^{KO/G373R}* P140 mice (*Figure 7A*). Moreover, the levels of LAMP2 are also reduced in the Purkinje cells of these injected mice (*Figure 7B*). Hence, injection of the gene therapy construct restores GlcCer levels and alleviates lysosome expansion 150 days after the uninjected mice died, clearly showing beneficial effects.

Next, we assessed the levels of Vps35 in injected and uninjected *Pla2g6^{KO/G373R}* mice. The levels of Vps35 are significantly reduced in the *Pla2g6^{KO/G373R}* mice, as anticipated (*Figure 7C*). Interestingly,

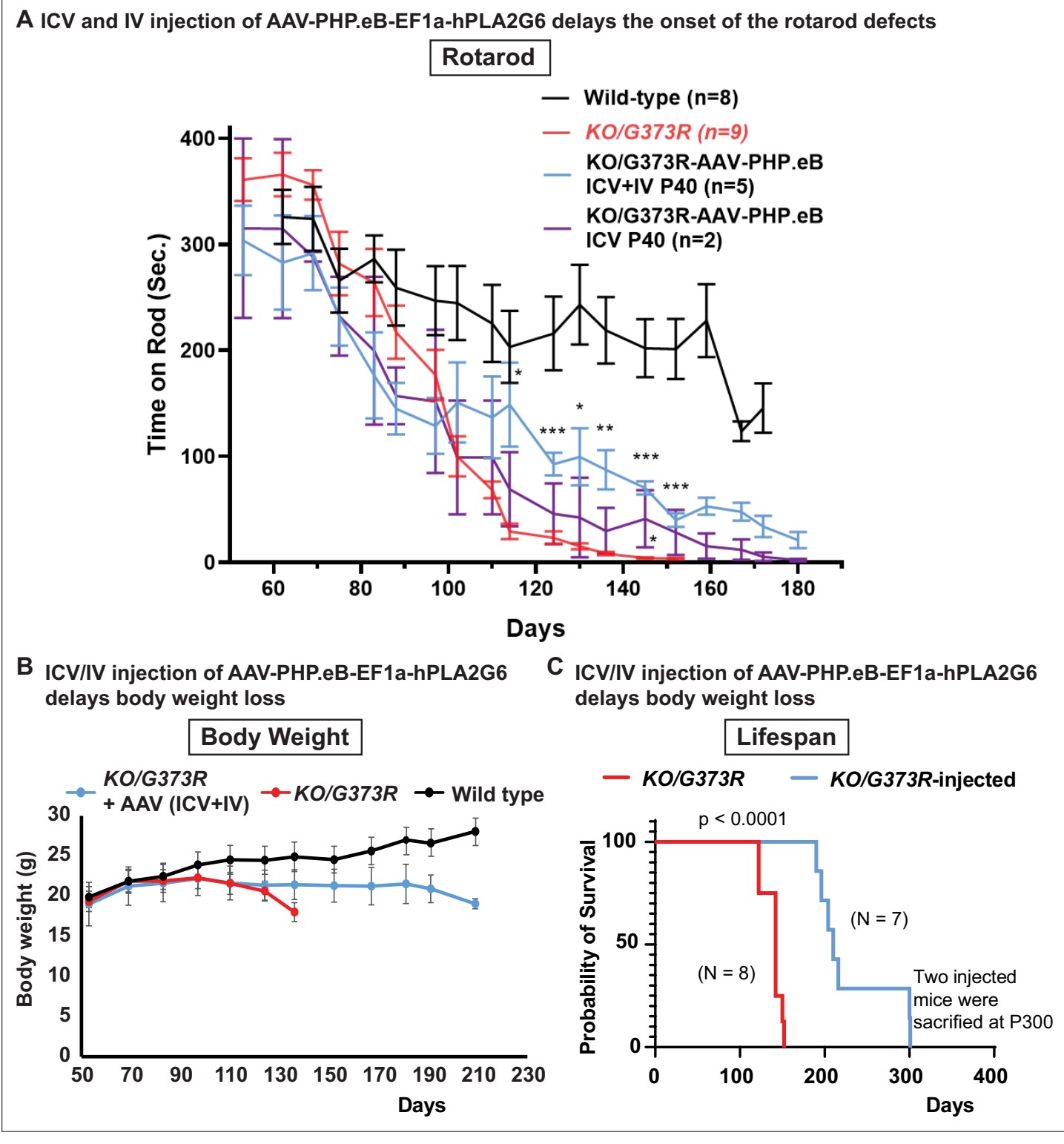

**Figure 6.** Pre-symptomatic injection of AAV-PHP.eB-PLA2G6 suppresses rotarod defect and prolongs lifespan in *Pla2g6^{KO/G373R}* INAD mice. (**A**) Rotarod defect in *Pla2g6^{KO/G373R}* mice (genotype: *Pla2g6^{KO/G373R}*) (n=6) are reduced by pre-symptomatic (**P40**) ICV + IV injection (n=5), but not ICV only (n=2). Wild-type (n=5); *Pla2g6^{KO/G373R}* (n=10). Rotarod performance of mice with the indicated genotypes was measured weekly. (**B**) Pre-symptomatic (**P40**) ICV + IV injection of AAV-PHP.eB-hPLA2G6 stabilizes body weight of *Pla2g6^{KO/G373R}* mice. Wild-type (n=5); *Pla2g6^{KO/G373R}* (n=6); *Pla2g6^{KO/G373R}* injected (n=3). (**C**) Pre-symptomatic (**P40**) ICV + IV injection of AAV-PHP.eB-PLA2G6 prolongs lifespan of the *Pla2g6^{KO/G373R}* mice. *Pla2g6^{KO/G373R}* (n=8); *Pla2g6^{KO/G373R}* injected (n=8). Error bars represent SEM. * $P<0.05$; ** $P<0.01$; *** $P<0.001$. All assays were conducted in blind of genotypes and treatments.

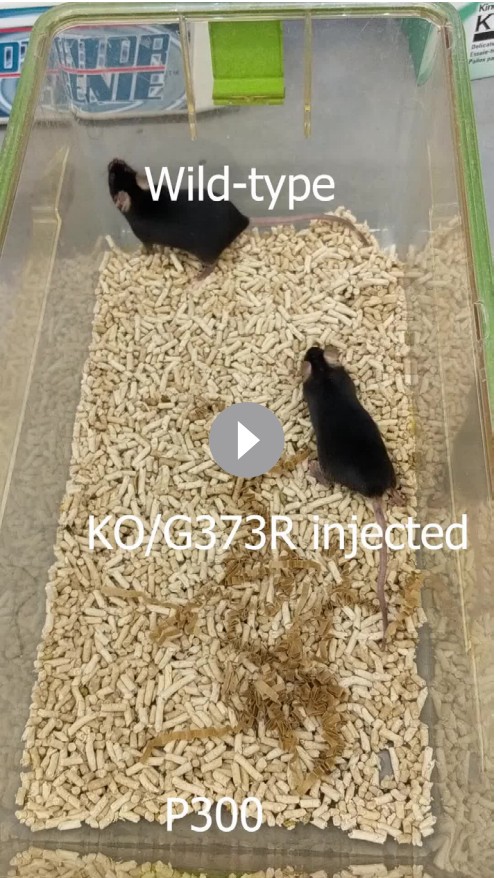

**Video 1.** The injected *Pla2g6^{KO/G373R}* mice (at ~P300) behaved relatively normal in their home cage. https://elifesciences.org/articles/82555/figures#video1

expression of human PLA2G6 not only restored the levels of Vps35, but also enhanced its expression (~2-fold of the wild-type mice) (**Figure 7C**). In summary, expression of human PLA2G6 using the gene therapy construct in *Pla2g6^{KO/G373R}* mice increases the Vps35 levels.

## Discussion

We previously documented that flies lacking *iPLA2-VIA*, the fly homolog of *PLA2G6*, have a dysfunctional retromer complex, accumulate ceramides, exhibit an expansion of lysosomes, and develop aberrant mitochondria (*Lin et al., 2018*). Here, we show that *PLA2G6* is highly expressed in iPSCs and NPCs but not in fibroblasts (**Figure 1A**). INAD patient-derived NPC and DA neurons that lack PLA2G6 also exhibit an impairment of the retromer, an expansion of the lysosomes, an increase in ceramides, and a disruption of mitochondrial morphology (**Figure 1** and **Figure 1—figure supplement 1**). We also observe similar defects in INAD mouse models (**Figure 2** and **Figure 2—figure supplement 1**). Hence, our data indicate that these defects are a root cause of INAD/PARK14.

Cerebellar atrophy is one of the earliest features shared by most INAD patients based on MRI studies (*Farina et al., 1999*). In *Pla2g6* knockout mice, cerebellar atrophy, as well as a loss of Purkinje neurons were observed in older animals (18 months) (*Zhao et al., 2011*). Moreover, mice with a knock-in of a variant identified in *PARK14* patients, exhibit an early onset loss of the substantia nigra and a loss of DA neurons (*Chiu et al., 2019*). These data suggest that loss of Purkinje neurons in the cerebellum and/or DA neurons in the substantia nigra are features that are shared by patients and mice that lack *Pla2g6*. We find that GlcCer is highly enriched in Purkinje and DA neurons in mutant *Pla2g6* mice (**Figure 2D**) and in INAD patient-derived DA neurons (**Figure 1B** and **Figure 1—figure supplement 1B**). We also observed a significant expansion of the lysosomes as well as mitochondrial defects in the mice DA neurons and Purkinje cells as well as in DA neurons derived from NPCs (**Figure 1C–F**, **Figure 2C**, and **Figure 2—figure supplement 1**), consistent with the observed lesions in patients.

In the past 2 years, *PLA2G6* has been shown to function as a key regulator of ferroptosis in cancerous cell lines and in placental trophoblasts. Elevated levels of reactive oxygen species (ROS) and iron lead to ROS-induced lipid peroxidation. This promotes ferroptosis (*Jiang et al., 2021*). Loss of *PLA2G6* in cancerous cell lines or placental trophoblasts promotes lipid peroxidation and ferroptosis (*Beharier et al., 2020*; *Chen et al., 2021*; *Kajiwara et al., 2022*; *Wang et al., 2022b*). An accumulation of peroxidated lipids at day 25 was also observed in adult brains of flies that carry a hypomorphic allele of *iPLA2-VIA* (*Kinghorn et al., 2015*). Moreover, *Pla2g6^{R748W/R748W}* mice that contain a PARK14 variant, exhibit an early impairment of rotarod performance, an elevation of ferroptotic death, and a loss of DA neurons in the midbrains in 7-month-old *Pla2g6^{R748W/R748W}* knock-in mice (*Sun et al., 2021*). Interestingly, ferroptosis is also observed in rotenone-infused rats as well as in α-synuclein-mutant *Snca^{A53T}* mice, suggesting that this pathway may be affected in different models of PD (*Sun et al., 2021*). However, in flies that lack *iPLA2-VIA*, we did not observe an elevation of iron or ROS in 15-day-old adults (these flies live a maximum of 30 days), yet these flies already exhibit a severe ceramide accumulation and lysosomal expansion. Similarly, the INAD mice exhibit a severe ceramide accumulation and lysosome expansion in Purkinje neurons and DA neurons prior to neuronal death. We therefore

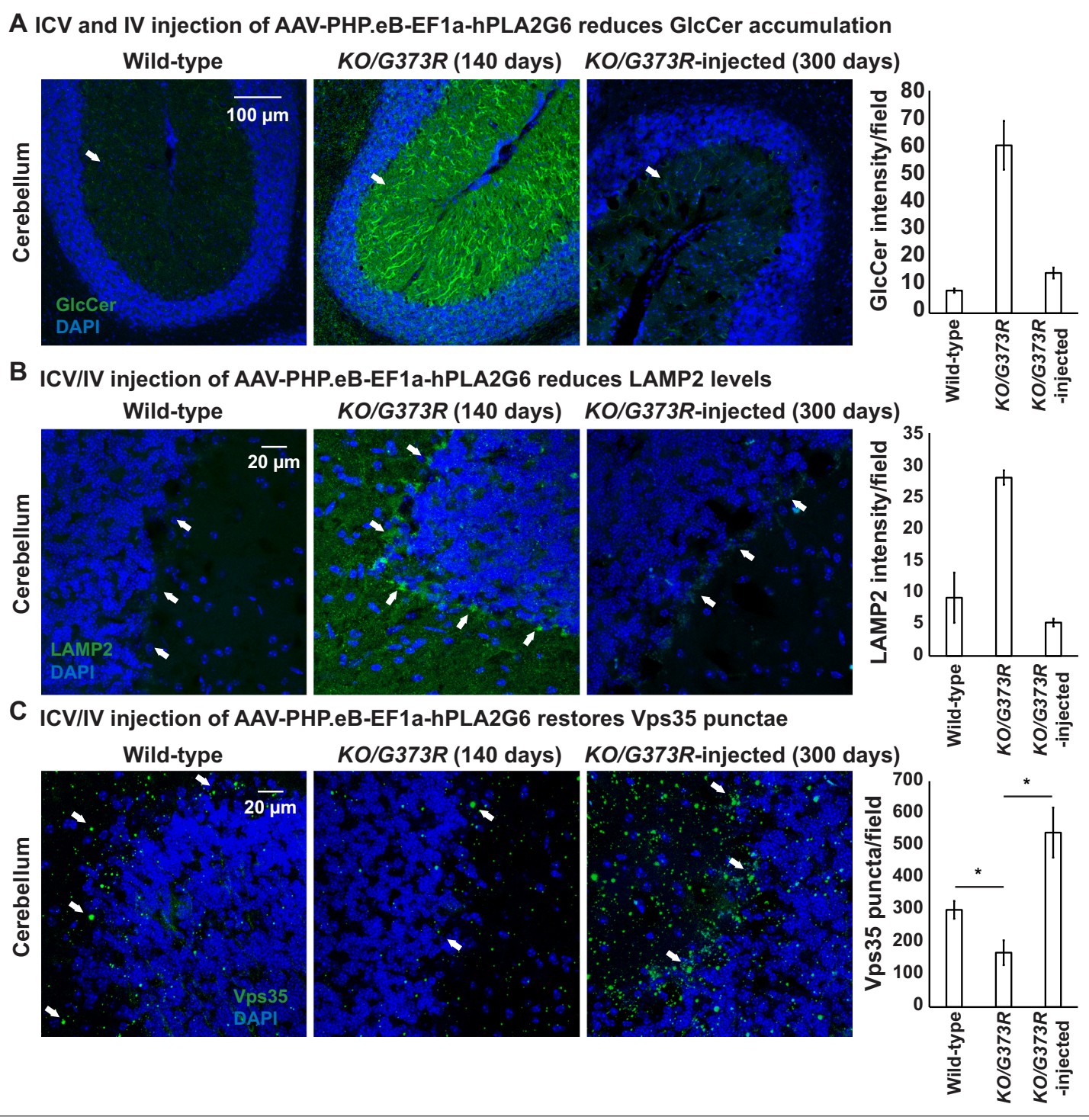

**Figure 7.** ICV and IV injection of AAV-PHP.eB-EF1a-PLA2G6 reduces GlcCer and LAMP2 accumulation and restores the number of Vps35 punctae. (**A**) Immunofluorescent staining of mouse cerebella of the indicated genotypes and treatments. GlcCer antibody (green; arrows) was used to assess the levels of GlcCer. Scale bar=100 μm. All assays were conducted blind for genotypes and treatments. (**B**) Immunofluorescent staining of mouse cerebella of the indicated genotypes and treatments. LAMP2 antibody (green; arrows) labels lysosomes in the Purkinje cells. Scale bar=20 μm. (**C**) Immunofluorescent staining of mouse cerebella of the indicated genotypes and treatments. Vps35 antibody (green; arrows). Scale bar=20 μm. Representative images are shown in this figure. DAPI (blue) labels nuclei. Representative images are shown in this figure. Quantifications are shown next to the images (n=3 for wild-type and *Pla2g6*[KO/G373R] mice; n=2 for *Pla2g6*[KO/G373R]-injected mice). Error bars represent SEM; *p<0.05; **p<0.01. ICV, intracerebroventricular; IV, intravenous.

argue that ceramide accumulation and lysosome expansion precede ferroptosis. Our data are also consistent with the observation that activation of lysosomes and autophagy promote ferroptosis (*Hou et al., 2016*; *Torii et al., 2016*). Hence, loss of *PLA2G6* may disrupt retromer function, and lead to impaired recycling of proteins and lipids which in turn causes lysosomal and autophagy defects that activate ferroptosis.

We screened drugs that affect sphingolipid metabolism, endolysosomal trafficking as well as drugs that are being explored in PD. Upon screening these drugs in flies, we identified drugs that had a positive impact on bang-sensitivity and subsequently screened them in NPCs derived from an INAD patient. We identified four drugs, Ambroxol, Desipramine, Azoramide, and Genistein that alleviate bang-sensitivity and ERG defects in INAD flies (*Figure 3B* and *Figure 3—figure supplement 1*). Moreover, these drugs also reduce LAMP2 and restore ATP levels in INAD patient-derived NPCs (*Figure 3C–E*). Upon oral uptake, Ambroxol is transported to lysosomes where it serves as a molecular chaperone for β-glucocerebrosidase (*Magalhaes et al., 2018*), the enzyme encoded by *GBA*, associated with Gaucher Disease and PD (*Sidransky, 2004*; *Wong et al., 2004*). Ambroxol promotes β-glucocerebrosidase activity in the lysosomes to reduce the levels of GlcCer (*Magalhaes et al., 2018*). Given that loss of *Pla2g6* leads to a robust accumulation of GlcCer in Purkinje neurons and in DA neurons, Ambroxol may promote the degradation of GlcCer and reduce the toxicity of GlcCer accumulation, consistent with our model. However, it should not affect the elevation of other ceramides and hence may only partially suppress the phenotypes associated with elevated ceramides. Indeed, RNAi knockdown of *lace*, the rate-limiting enzyme that synthesizes ceramides, strongly impairs ceramide synthesis, and has the most potent suppressive effect as it affects all ceramides (*Lin et al., 2018*).

Interestingly, Miglustat and Ibiglustat, two inhibitors of UGCG UDP-glucose ceramide glucosyltransferase that are being used to reduce GlcCer levels in Gaucher disease patients, did not reduce the bang-sensitivity in INAD flies (*Figure 3B*). UGCG UDP-glucose ceramide glucosyltransferase is the enzyme that generates GlcCer. Blocking its activity reduces GlcCer levels but does not affect the other ceramides. We argue that this may lead to an elevation of other ceramides, and these drugs may not be effective given that many other ceramides are elevated in INAD. Indeed, Desipramine, a tricyclic antidepressant that is transported to the lysosomes where it functions as an acidic sphingomyelinase inhibitor to suppress the overall ceramide synthesis (*Jenkins et al., 2011*), showed a significant rescue effect (*Figure 3*). Hence, based on our drug screen, we identify two drugs, Ambroxol and Desipramine, that reduce the levels of ceramides and lysosomal defects (*Lin et al., 2018*). These data are consistent with our model that increased ceramide levels contribute to the pathogenesis of INAD and *PARK14*.

Azoramide, a drug tested for parkinsonism, promotes protein folding and secretion without inducing ER stress in the endoplasmic reticulum (*Fu et al., 2015*). It reduces ER stress, abnormal calcium homeostasis, mitochondrial dysfunction, elevated ROS as well as cell death in PARK14 patient-derived DA neurons (*Ke et al., 2020*). Azoramide reduces bang-sensitivity in flies and reduces the lysosomal expansion in INAD patient-derived NPCs (*Figure 3*). Azoramide promotes protein folding and hence may reduce the levels of misfolded proteins and alleviate lysosomal stress (*Jackson and Hewitt, 2016*). Finally, Genistein is an isoflavone naturally found in soy products. It exhibits neuroprotective effects in DA neurons in the MPTP-induced mouse model of Parkinson's disease (*Liu et al., 2008*). It has been shown to promote lysosomal biogenesis by activating the transcription factor EB (TFEB) (*Moskot et al., 2014*). Taken together, the identification of Azoramide and Genistein to alleviate neurodegeneration in INAD models indicates that lysosomal defects are indeed key contributors to the pathogenesis of INAD.

We previously showed that whole-body expression of human *PLA2G6* cDNA in flies that lack *iPLA2-VIA* completely rescued the neurodegenerative defects as well as lifespan, whereas neuronal expression strongly suppressed the neuronal phenotypes but did not extend lifespan (*Lin et al., 2018*). To assess the effect of expressing PLA2G6 in INAD mice and patient NPCs, we created two AAV-based vectors, *AAV-EF1a-EGFP* and *AAV-EF1a-PLA2G6*, which express EGFP or human PLA2G6 under the control of a ubiquitous promoter (*EF1a*) in human NPCs and mice. After delivery using ICV and IV into mice, the EGFP is broadly expressed in the nervous system as well as in other organs including the heart and liver (*Figure 4—figure supplement 1*). Upon delivery of *AAV-EF1a-PLA2G6* to patient-derived NPCs, the construct expresses low levels of PLA2G6 (10–20% of the endogenous levels). However, this is sufficient to partially alleviate the defects including Vps35 levels, lysosomal

expansion and mitochondrial morphological and functional abnormalities in NPCs (*Figures 4 and 5*). Moreover, delivery of *AAV-EF1a-PLA2G6* into *Pla2g6*$^{KO/G373R}$ mice also delays the onset of rotarod defect, helps to sustain body weight, and prolongs lifespan (*Figure 6*). Importantly, it also reduces the levels of GlcCer and LAMP2, and increases the levels of Vps35 in *Pla2g6*$^{KO/G373R}$ INAD mice (*Figure 7*). These proof-of-concept data are important because they indicate that the defects caused by a loss of *PLA2G6* can be delayed by low levels of *PLA2G6* expression and reversed in NPCs. However, the efficiency of delivery and expression need to be improved, possibly by testing other serotypes of AAV or injecting more virus.

In summary, the accumulation of ceramides, expansion of lysosomes, and disruption of mitochondria are key phenotypes that are associated with the loss of *PLA2G6* in flies, mice, and human cells. The defects are obvious in mouse DA neurons and Purkinje cells, two cell populations that have been implicated in INAD/PARK14. We also identified four drugs that suppress the ceramide levels or promote lysosome functions and that alleviate the defects caused by loss of *PLA2G6* in INAD flies and in INAD patient-derived NPCs. These data, combined with previous data, provide compelling evidence that endolysosmal trafficking defects, expansion of lysosomes, elevation of ceramides, and disruption of mitochondria are at the root of the pathogenesis of INAD and PARK14. Finally, we report encouraging data for a proof-of-concept trial to test the efficiency of a gene therapy approach. We argue that combining a drug and gene therapy approach will provide an avenue to significantly improve the quality of life of INAD/PARK14 patients.

# Materials and methods

## *Drosophila* and drug treatment

Mixed genders of flies (approximately 50% male and 50% female) were used for all experiments. Flies were raised on molasses-based food at 25°C in constant darkness. The genotypes of the flies used is *y w;; iPLA2-VIA*$^{Δ174}$ (*Lin et al., 2018*). All drugs were added freshly to regular fly food at the indicated concentration. The flies were transferred to fresh food with or without the drugs every 3 days.

## *Drosophila* behavioral assay

To perform bang-sensitive paralytic assays, five adult flies were tested per vial. The flies were vortexed at maximum speed for 15 s and the time required for flies to stand on their feet was counted. At least 60 flies were tested per data point.

## Fibroblasts from INAD patients and their parents

Human skin fibroblasts from Families 1 and 2 were a gift from Dr. Bénédicte Heron (Neurologie Pédiatrique Hôpital Trousseau) and Dr. Young-Hui Jiang (Duke University), respectively. The control skin fibroblasts (Cat. #GM23815) were purchased from Coriell Institute.

## Human iPSC culture

Human iPSCs (hiPSCs) maintained on Cultrex (CTX Cat# 3434-010-02, R&D Systems) in PSC Freedom Media (FRD1, ThermoFisher Custom) were passaged every 4–5 days using Stem-Accutase (Cat# A11105-01, Life Technologies) in the presence of 1 μM Thiazovivin (THZ Cat# SML1045-25 MG, Sigma-Aldrich).

## CRISPR/Cas9-mediated gene editing

### Transfection

$4×10^5$ Stem-Accutase dissociated hiPSCs were plated onto a CTX pre-coated 24 wells in FRD1 containing THZ. Cells were transfected directly after passaging with the transfection reaction. An RNP complex was formed by incubating 18 pM Alt-R S.p. Cas9 Nuclease V3 (Cat# 1081058 IDT Technologies) with 18 pM PLA2G6 Alt-R CRISPR-Cas9 sgRNA (GTGAGTTCCTGGGGTTGACC, IDT) for 5 min at RT. The RNP complex was then incubated for 15–20 min at RT with 120 μM Alt-R HDR Oligos (tcca atccgagacgtggggggagtgaaaggagagaagtatgttcccgctgagcatcacccaccggaatccactctgtgagttcctggggttga ccaAgacgcagtcccaggtgcggttggggagtgttc, IDT Technologies), 2 μl Lipofectamine Stem (Thermo Fisher Scientific, STEM00008), and 40 μl Opti-MEM (Cat# 31985062, Thermo Fisher Scientific).

## Monoclonalization

Transfected iPSCs were single cell sorted into 96-well plates using a Benchtop Microfluidic Cell Sorter (Nanocollect). Plates were fed daily with FRD1 and scanned every night on a Celigo Image Cytometer (Nexcelom Bioscience). After 10 days, monoclonal colonies were consolidated and passaged into a new 96-well plate. Wells were passaged when reaching 80–100% confluency for freeze backs and sequencing analysis.

## Sanger sequencing of monoclonal wells

About 30 µl of QuickExtract DNA Extraction (Lucigen, QE09050) was added to $5.0 \times 10^4$ pelleted iPSC, resuspended, and incubated for 15 min at 65°C. Quick extract lysate template was prepared by positing $5.0 \times 10^4$ cells into a 96-well hard-shell PCR plate (Bio-Rad). The PCR for Sanger sequencing was performed by using 2 µl of quick extracted gDNA in a 25 µl PCR reaction using AmpliTaq Gold 360 and PLA2G60 primer pairs (fwd: gccgcctggtcaataccttc, rev: acccctcagacagagactcaa). The amplicon was sent for Sanger sequencing subsequently.

## Quality control measures

iPSCs were expanded via automation on the NYSCF Global Stem Cell Array platform for further quality control assays and then frozen into barcoded Matrix tubes in Synth-a-freeze Cryopreservation Media at R500k cells/vial.

All iPSC lines undergo rigorous quality control that includes a sterility check, mycoplasma testing, viability, karyotyping via Illumina Global Screening Array, SNP ID fingerprinting via Fluidigm SNPTrace, pluripotency, and embryoid body scorecard assays via Nanostring. iPSCs were maintained using Freedom (Thermo Fisher Scientific, custom) media.

# Generation of the isogenic INAD patient-derived iPSC, NPCs, and DA neurons

Undifferentiated iPSCs were grown following standard protocols in StemFlex medium on laminin521 substrate (BioLamina) as described previously (*Ruzo et al., 2018*). iPSCs were differentiated to ventral midbrain NPCs and DA neurons following (*Kim et al., 2021*) with the following modifications. Neural induction was initiated at day 0 by passaging iPSC, single cell seeding at 1 M cells/ml in 2.5 M/well of over ultralow adherence plastic 6wp (Corning, Cat #3471vendor) in neural induction medium (base medium changed to AaDMEMEMM/F12:Neurobasalbobasla 50:50, with N2, B27 without RA, RhoK inhibitor Y27632 10 µM (first 2 days), 100 nM LDNn, 10 µM SB-431542 and, substituting 1 mM SAG3.1 and 1 mM Prurmorprphhpamione (R&D) in place of SHH protein), and 1 µM CHIR99021 (R&D). At days 4–7, CHIR99021 was increased to 6 µM, and from days 8 to 11 was decreased to 3 µM and 0.2 mM AA (Sigma-Aldrich), 0.2 mM dbcamp (Sigma-Aldrich), 10 µM DAPT (Tocris), and 1 ng/ml TGFb3 (R&D) 0 were added from day 10 onward. EBs were dissociated with accutase at day 16, and reseeded at 0.8Mcells/cm² on polyornithine laminin (1 mg/ml in pH 8.4 borate buffer followed by 10 µg/ml natural mouse laminin in DMEM/F12) coated TC plastic for continued differentiation and day 22 fixation and staining for positional markers, or frozen in 2× FM (Millipore Sigma; ES-002-10F). Live cultures or thawed cells were either expanded as floor plate progenitors (following Brundin approach; Floor Plate Progenitor Kit expansion protocol; Thermo Fisher Scientific; A3165801) or terminally differentiated to DA neurons with all prior factors except in neurobasal base medium after day 25.

## Western blotting

Cells were homogenized in 1% NP40 lysis buffer (20 mM HEPES pH7.5, 150 mM NaCl, 1% NP-40, 10% Glycerol, and Roche protease inhibitor mix) on ice. Tissue or cell debris were removed by centrifugation. Isolated lysates were loaded into 10% gels, separated by SDS-PAGE, and transferred to nitrocellulose membranes (Bio-Rad). Primary antibodies used in this study were as follows: mouse anti-PLA2G6 antibody (Santa Cruz Biotechnology; sc376563), rabbit anti-PLA2G6 antibody (Sigma-Aldrich; SAB4200129), rabbit anti-PLA2G6 antibody (Sigma-Aldrich; HPA001171), mouse anti-Actin (ICN691001, Thermo Fisher Scientific), and rabbit anti-LAMP2 antibody (Abcam; ab18528).

## Immunofluorescence staining

The cultured cells were fixed with 4% paraformaldehyde (PFA) in 1× PBS at 4°C for 30 min. The fixed cells were permeabilized in 0.1% Triton X-100 in 1× PBS for at least 15 min at room temperature. For mouse tissues, the mouse was deeply anesthetized using isoflurane, and perfused intracardially with 1× PBS and 4% PFA. Brains were dissected and post-fixed in 4% PFA/PBS overnight at 4°C. The next day, tissues were cryoprotected in a 20% sucrose/PBS solution at 4°C for 1 day, followed by a 30% sucrose/PBS solution at 4°C for one more day. Tissues were then embedded and frozen in OCT and cryosectioned using a cryostat (Leica CM1860) at 25–40 µm. The fixed mouse tissues were permeabilized in 0.1% Triton X-100 in 1× PBS for at least 15 min at room temperature. The following antibodies were used for the immunofluorescence staining: chicken anti-Tyrosine Hydroxylase antibody (Abcam; ab76442; RRID:AB_1524535); rabbit anti-LAMP2 antibody (Abcam; ab18528; RRID:AB_775981); mouse anti-TOMM20 antibody (Abcam; ab56783; RRID:AB_945896); rabbit anti-GlcCer (Glycobiotech; RAS_0010); mouse anti-ATP5a (Abcam; ab14748; RRID:AB_301447); mouse anti-LAMP1 (Abcam; ab25630; RRID:AB_470708); goat anti-Vps35 (Abcam; ab10099; RRID: AB_296841); mouse anti-GFP FITC conjugated (Santa Cruz Biotechnology; sc9996; RRID:AB_627695); NESTIN (Millipore; ABD69; RRID:AB_2744681); FOXA2 (R&D; AF2400; RRID:AB_2294104); LMX1A (Millipore; AB10533; RRID:AB_10805970); OTX2 (R&D; AF1979; RRID:AB_2157172); FOXG1 (Takara; ABM227); and Alexa 488-, Cy3-, or Cy5-conjugated secondary antibodies (111-545-144, 111-585-003, and 111-175-144); Jackson ImmunoResearch Labs; or Alexa 488, 555, 647 conjugated anti-rabbit, mouse, and goat (Thermo Fisher Scientific). All the confocal images were acquired with a Model SP8 confocal microscope (Leica), with the exception of vmNPC characterization and initial Da neuron characterization, acquired with a Phenix spinning disk confocal HTS microscope (PE) and LSM700 Confocal microscope (Zeiss), respectively. Confocal images were processed using ImageJ and Photoshop (Adobe).

## Molecular cloning of the gene therapy constructs

Construction of pAAV-EF1a-PLA2G6: PCR amplified fragments of the full-length human PLA2G6 cDNA (NM_001349864.2) containing XbaI (5′) and EcoRV (3′) were digested with XbaI and EcoRV. To generate the vector backbone, pAAV-EF1a-CVS-G-WPRE-pGHpA (Addgene; 67528) was digested with XbaI and EcoRV and gel eluted to remove the CVS-G. The digested PCR fragments were ligated in the digested vector backbone to generate pAAV-EF1a-PLA2G6-WPRE-pGHpA. The pGHpA was replaced by SV40pA to reduce the total size of the construct. Similarly, the pAAV-EF1a-EGFP was constructed following the same method except the EGFP cDNA was used. Lenti-CMV-PLA2G6 was purchased from Horizon Discovery-Precision LentiORF collection (OHS5897-202617053).

## AAV-PHP.eB package

A plasmid DNA cocktail solution was produced by combining the pAAV transgene (pAAV-EF1a-PLA2G6 or pAAV-EF1a-EGFP), rep/cap serotype PHP.eB, and AdΔF6 helper plasmids and transfected into HEK293T cells with iMFectin poly transfection reagent – GenDEPOT. A total of 80×15 cm$^2$ dishes were overlaid with the DNA cocktail solution and were allowed to incubate for 4 hr before adding fresh media. Three days post-transfection, dishes were imaged, harvested, and digested. The collected cell viral lysate was centrifuged, supernatant was transferred to a new tube and digested separately. The cell pellet was resuspended in TMN (Tris-HCl cell suspension buffer) and cell-associated AAV was recovered by cell lysis treatment with 5% sodium deoxycholate, DNase, RNase, and three subsequent freeze/thaw cycles. Media-secreted AAV was precipitated in a 40% polyethylene glycol solution. Digest CVL was centrifuged to remove cell debris and cleared supernatant was transferred to a new tube. PEG precipitated supernatant was centrifuged, the resulting pellet was resuspended in HBS (HEPES Buffered Saline) solution. After digestion and precipitation, cleared cell viral lysate and media secreted AAV were combined and purified on a discontinuous iodixanol gradient. The band corresponding to purified AAV was extracted and diluted in DPBS supplemented with 0.001% Pluronic F-68. Viral diluent was then concentrated in Amicon centrifugation filtration units (100,000 MW) to the desired volume. Concentrated AAV was first diluted 1:100 and then serially diluted 10-fold to yield AAV dilutions of 0.01, 0.001, 0.0001, and 0.0001. The titer of the AAV vectors was quantified with the primers corresponding to WPRE and probed against a GVC in-house standard. The AAV-PHP.eB was packaged by the Gene Vector Core at Baylor College of Medicine.

## Mouse house keeping

All experimental animals were treated in compliance with the United States Department of Health and Human Services and the Baylor College of Medicine IACUC guidelines. Mice were reared in 12-hr light-dark cycles with access to food and water ad libitum. The mouse lines used here include the *Pla2g6* complete knockout line (genotype: *Pla2g6$^{KO/KO}$*) (*Bao et al., 2004*) and the *Pla2g6$^{G373R}$* point mutation mouse line (Genotype: *PLA2G6$^{G373R/G373R}$*) (*Wada et al., 2009*). The PLA2G6 complete knockout line was a donation from Dr. Sasanka Ramanadham at the University of Alabama at Birmingham. Cryopreserved sperm of the *Pla2g6$^{G373R}$* mice (No. BRRC04196) was purchased from RIKEN BioResource Research Center. The cryopreserved sperms were used to in vitro fertilize female C57BL/6 mice to retrieve the line. The in vitro fertilization was performed by the GERM Core at Baylor College of Medicine. After the *Pla2g6$^{G373R}$* point mutation mice were retrieved, we mated *P la2g6$^{KO/+}$* with *P la2g6$^{G373R/+}$* mouse to generate transheterozygous mice (Genotype: *P la2g6$^{KO/G373R}$*). Mice were genotyped by standard PCR using isolated tail DNA as template.

## Mouse stereotaxic CNS injection

Preoperatively, adult mice were weighed and given meloxicam 30 min prior to surgery with a 30-gauge needle to minimize discomfort. Animals were anesthetized preoperatively with isoflurane. Following anesthesia, animals were checked for pedal withdrawal reflex; if absent, the animals were transferred to stereotaxic apparatus and maintained under anesthesia using volatilized isoflurane (1–3% depending on physiological state of the animal, which was continuously monitored by response to tail/toe pinch). Isoflurane was diluted with pressurized oxygen using an anesthetic vaporizer system that allows precise adjustment of isoflurane and gas flow to the animal. Surgical site was prepared by shaving hair, administering depilatory cream and three times betadine surgical scrub, three times ethanol wipe. A short incision (0.5–2 cm) was made over the skull using small surgical scissors. Following the incision, a small burr hole craniotomy (<1 mm in diameter) was made using a dental drill 0.5 µl of AAV particles were injected into the third ventricle (coordinates from Bregma in mm: ML 1, AP –0.3, DV –2.3) using a 33-gauge Hamilton syringe. Post-injection, the scalp was sutured using surgical nylon monofilament (Ethicon Cat# 1689G) in a simple interrupted pattern for skin-to-skin closure. After 7 days, mice were briefly anesthetized with isoflurane and sutures were gently removed using small scissors and forceps.

## Mouse tail vein injections

Random selected mice were placed in restrainer to immobilize and have easy access to the tail. Tail was soaked in warm water for 10–15 s to cause vasodilation (enlargement) of the vein and was swabbed with a gauze dampened with alcohol to increase the visibility of the vein. One of the two lateral tail veins was located in the middle third of the tail. With the bevel of the needle facing upward and the needle almost parallel to the vein, the needle was slid into the tail vein. By gently applying negative pressure to the plunger and observing a flash of blood in the hub vein penetration was confirmed. Slowly the plunger was pressed to inject AAV-containing solution into the vein. The needle was removed from the vein and slight pressure was applied to the puncture site with a dry piece of gauze until the bleeding has stopped. The animal was removed from its restrainer and placed in the cage. The animal was monitored for 5–10 min to ensure hemostasis and that the bleeding has stopped.

## Mouse rotarod assay

We measured the time (latency) a mouse takes to fall off the rod under continuous acceleration. On the day of testing, mice were kept in their home cages and acclimated to the testing room for at least 15 min (acclimation phase). The apparatus was set to accelerating mode from 5 to 40 rpm in 300 s. We record the latency at which each mouse falls off the rod. The first trial is the training period. During the training period, the mice were tested three times separated by 15 min intervals in a day and the trials were repeated for 3 consecutive days. After the training period, the mice were tested once a week. At the end of the rotarod assay, we weighted each mouse every other week. All the mice were tested in blind of genotype and treatments.

## Quantification and statistical analysis

For fly experiments, sample sizes are stated in the figure legends. The drug treatments were performed for three biological replicates. For all cell experiments, the studies were conducted in parallel with

vehicle controls in the neighboring well for at least three biological replicates. All mouse experiments were conducted in the number of technical replicates indicated in the figure legends. Error bars are shown as standard error of the mean (SEM). The mouse survival data sets (*Figure 2B* and *Figure 5C*) were organized and analyzed in GraphPad Prism 9.4.0 using Mantel-Cox and Gehan-Breslow-Wilcoxon tests. All other data sets from flies and cells were organized and analyzed in Microsoft excel 2010 using two-tailed Student's t test. The criteria for significance are NS: not significant; $p > 0.05$; *$p < 0.05$; **$p < 0.01$, and ***$p < 0.001$.

## Lipid extraction

Lipids were extracted from the method as described previously from fibroblast cells and cerebellum tissue samples (PMID 26090945, 35857503). d18:1/16:0-d3-GlcCer, deuterated ceramide LIPIDOMIX (mixture of d18:1-d7/16:0-, d18:1-d7/18:0-, d18:1-d7/24:0-, and d18:1-d7/24:1-Cer), d18:1-d7 sphingosine (So), and d18:0-d7-sphinganine (Sa), all purchased from Avanti Polar Lipids, Inc, were used as internal standards (IS).

For fibroblasts, IS mixture was added equally to cell pellets equivalent to $1 \times 10^6$ cells with 500 μl 1-butanol:methanol (1:1, v/v) as extraction solvent. Samples were tip-sonicated on ice, followed by water bath sonication for 60 min at room temperature and centrifugation at 13,000×*g* for 10 min. The supernatant was collected and dried using nitrogen air and reconstituted in chloroform:methanol (1:4, v/v) and centrifuged at 13,000×*g* for 5 min to remove any insoluble salt.

For tissue samples, all tissue samples were hard-frozen in liquid nitrogen, followed by cryopulverization. Powdered samples were sonicated in water bath for 60 min and centrifuged at 13,000×*g* for 10 min. The supernatant was collected and dried using nitrogen air and reconstituted in chloroform:methanol (1:4, v/v) and centrifuged at 13,000×*g* for 5 min to remove any insoluble salt.

## Targeted analysis of lipids using LC-MS/MS

Vanquish Horizon UHPLC (Thermo Fisher Scientific, Waltham, MA) was coupled to Thermo Altis triple quadrupole (Thermo Fisher Scientific, Waltham, MA) for targeted analysis of lipids. Sphingosine, sphinganine, hexosylsphingosine, hexosylceramides, and ceramides were analyzed on a Atlantis T3 (1.0×50 mm², 3 μm) under a binary gradient of mobile phase A (water:acetonitrile, 9:1, v/v) and mobile phase B (isopropanol:methanol:acetonitrile, 7:2:1, v/v/v) with 0.1% formic acid at the flow rate of 150 μl/min. Mobile phase B was increased from 10% to 30% over 1 min, 40% over 1 min, 70% over 1.5 min, 72.5% over 3.5 min, and 95% over 3.5 min. After maintaining the mobile phase B at 95% for 1.5 min, it was ramped down to 10% over 0.1 min and re-equilibrated at 10% mobile phase B for 3 min. Scheduled multiple reaction monitoring of targeted lipids were acquired with Q1 resolution at 0.4 FWHM, Q3 resolution at 0.7 FWHM, spray voltage 3.5 kV, sheath gas at 35, auxiliary gas at 5, and sweep gas at 1 in positive ion mode. Quantifier ion of *m/z* 264.25 and qualifier ion of *m/z* 282.25 were used for Cer species containing d18:1 long-chain base. For Cer species containing d18:0 long-chain base, *m/z* 284.25 as quantifier ion and *m/z* 266.25 as qualifier ion were used. For HexCer species, *m/z* 264.25 as quantifier ion and *m/z* equivalent to loss of hexose ring from the precursor ion was used as qualifier ion. Product ions of *m/z* 282.25 for So and hexosylsphingosine, and *m/z* 284.25 for Sa were utilized as quantifier ion. The analytical column was kept at 40°C throughout the analysis.

All lipids were quantified by calculating the peak areas and ratio of analyzed lipids over the IS's peak areas were calculated. Ratio of lipids against IS were normalized by the protein amount measured the samples using BCA protein assay. The total level of ceramides and hexosylceramides were calculated by adding the normalized peak areas ratios of individual species within each class.

## Acknowledgements

We thank Dayna McKenzie and Drs. Josephine Wesely and Brigham Hartley from the New York Stem Cell Foundation Research Institute for their contribution to genetically correct the INAD patient-derived iPSCs. The authors thank Drs. Jin Xu for very insightful comments. The authors thank the Gene Vector Core at Baylor College of Medicine for packaging the AAV-PHP.eB virus. The authors thank the GERM Core at Baylor College of Medicine for the in vitro fertilization to retrieve the *Pla2g6*[G373R] mice. The authors are grateful for the generous gift of skin fibroblasts from Dr. Bénédicte Heron (Neurologie Pédiatrique Hôpital Trousseau) and Dr. Young-Hui Jiang (Duke University). This project was supported in part by Baylor College of Medicine IDDRC Grant number P50HD103555 from the Eunice Kennedy

Shriver National Institute of Child Health and Human Development for use of the Microscopy Core facilities. The authors acknowledge the Shan and Lee-Jun Wong Fellowship to GL. The authors thank the INADcure Foundation for support to GL. HJB and the NYSCF. This project was also supported by the Huffington Foundation and the Jan and Dan Duncan Neurological Research Institute-Chair in Neurogenetics of TCH to HJB.

## Additional information

### Competing interests

Hugo J Bellen: Reviewing editor, eLife. The other authors declare that no competing interests exist.

### Funding

| Funder | Grant reference number | Author |
|---|---|---|
| Baylor College of Medicine | P50HD103555 | Hugo J Bellen |
| Huffington Foundation | | Hugo J Bellen |
| Shan and Lee-Jun Wong Fellowship | | Guang Lin |
| INADcure Foundation | | Guang Lin Hugo J Bellen |
| Jan and Dan Duncan Neurological Research Institute- Chair in Neurogenetics | | Hugo J Bellen |

The funders had no role in study design, data collection and interpretation, or the decision to submit the work for publication.

### Author contributions

Guang Lin, Conceptualization, Resources, Data curation, Software, Formal analysis, Supervision, Funding acquisition, Validation, Investigation, Visualization, Methodology, Writing - original draft, Project administration, Writing – review and editing; Burak Tepe, Regine C Tipon, Data curation, Formal analysis, Investigation, Methodology, Writing – review and editing; Geoff McGrane, Resources, Project administration, Writing – review and editing; Gist Croft, Formal analysis, Supervision, Investigation, Methodology, Writing – review and editing; Leena Panwala, Amanda Hope, Resources, Funding acquisition, Project administration, Writing – review and editing; Agnes JH Liang, Data curation, Formal analysis, Methodology; Zhongyuan Zuo, Seul Kee Byeon, Lily Wang, Akhilesh Pandey, Data curation; Hugo J Bellen, Conceptualization, Resources, Formal analysis, Supervision, Funding acquisition, Investigation, Methodology, Project administration, Writing – review and editing

### Author ORCIDs

Guang Lin http://orcid.org/0000-0001-5594-3397
Burak Tepe http://orcid.org/0000-0003-4371-2502
Regine C Tipon http://orcid.org/0000-0002-5473-7353
Akhilesh Pandey http://orcid.org/0000-0001-9943-6127
Hugo J Bellen http://orcid.org/0000-0001-5992-5989

### Ethics

All experimental animals were treated in compliance with the United States Department of Health and Human Services and the Baylor College of Medicine IACUC guidelines. Protocol (AN-5596).

### Decision letter and Author response

Decision letter https://doi.org/10.7554/eLife.82555.sa1
Author response https://doi.org/10.7554/eLife.82555.sa2

## Additional files

### Supplementary files
• MDAR checklist

### Data availability
All data generated or analyzed during this study are included in the manuscript and supporting file; Source Data files have been provided for Figures 1, 3, 4 and Suppl. Figure 1.

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
