## [Editor Report]

This important study is of significant interest to those studying neurodegeneration, demonstrating key pathologies in PLA2G6-associated disease in both patient-derived neuronal models and a novel trans heterozygote mouse model. It identifies a number of possible compounds that could potentially be re-purposed for therapeutic use in PLA2G6-associated neurodegeneration and provides a proof-of-principle in mouse that gene therapy with human PLA2G6 can rescue defects in PLA2G6 deficiency. The data are solid and convincing, and will stimulate future work to elucidate the precise molecular mechanisms at play.

---

## [Decision Letter]

**Decision letter after peer review:**

Thank you for submitting your article "Exploring therapeutic strategies for Infantile Neuronal Axonal Dystrophy (INAD/PARK14)" for consideration by *eLife*. Your article has been reviewed by 2 peer reviewers, and the evaluation has been overseen by Suzanne Pfeffer as the Senior Editor. The reviewers have opted to remain anonymous.

You will see below that the reviewers felt that further characterization of the diverse aspects you cover would add greatly to the overall impact of this study. We will leave it to you to be guided by these comments and add additional information to bolster the story and address the comments below.

*Reviewer #1 (Recommendations for the authors):*

Points to address:

iPSC/NPC work

– While the patient IPSC-derived neuronal cell data is interesting and confirms previous findings in *Drosophila* models of PLA2G6 knockout, it does not fully recapitulate all the pathologies previously reported by the authors in the INAD flies. The fly work led to the hypothesis that loss of PLA2G6 leads to loss of normal retromer function with consequent sphingomyelin/CPE mis trafficking and endolysosomal dysfunction, resulting in elevated ceramide levels. Does loss of PLA2G6 in the patient-derived neuronal cells lead to retromer dysfunction and increased ceramide levels? Are levels of Vps35 or Vps26 altered in the patient derived cells? It is interesting that GlcCer levels are up, but it would be useful to look specifically at different ceramide species. The reason for this is that the drugs previously used to rescue neurodegeneration were focused on reducing ceramide levels and not GlcCer. The significance of elevated GlcCer is not clear, and indeed drugs used in Gaucher's disease patients to specifically lower GlcCer were not successful in rescuing the PLAn phenotypes in this study.

– Figure 1E: It is not clear how the mitochondrial fragmentation phenotypes were quantified. How many cell/images were quantified? Was any quantification performed for the LAMP2 immunostaining? Although it looks like there is an obvious difference between patient and control NPCs, it should at least be stated that the images are representative.

– The current study demonstrates mitochondrial morphological abnormalities (with fragmentation). Is there also evidence of mitochondrial dysfunction in the patient-derived cells?

Mouse work

– Again, as with the patient-derived cell data, the mouse work is of significant interest but would benefit from exploration with regards to retromer function, ceramide levels and mitochondrial function in view of the previous hypothesis put forward from the fly work. The authors state that there is ceramide accumulation in the introduction summary and in the summary at the end of the mouse results – strictly speaking accumulation of GlcCer has been shown.

Drug screening

– The general feel of this work is that although it is of considerable interest, because of the potential for drug-repurposing for therapeutic purposes in PLAN, it has not been performed very rigorously. The candidate compounds have only been tested using one fly readout and a western blot in NPCs. Although statistically significant, the biological relevance of the degree of rescue of bang sensitivity is difficult to assess based on this data alone. This work would benefit from confirmatory studies of the molecular effectiveness of the drug. Ideally it would next be tested in the mouse models to show true efficacy at the lifespan and behavioural phenotype level. At the very least, validation of the drugs that were effective at rescuing bang-sensitivity require further validation in vivo in the fly model (e.g. ERG, photoreceptor loss suppression).

Gene therapy work

– While this work is of significant interest, the only cellular defect that is shown to be improved in the patient-derived NPCs is that of the mitochondrial fragmentation and LAMP2 expression. A reduction in ceramide levels or improvement in retromer function would support the efficacy at the cellular level. It is not clear how the fragmentation phenotype has been quantified for the NPCs (as in Figure 1E). The figure legend does not state that the images are representative. There is no indication of how many cells/images were quantified in the figure legend or in the methods.

– The rescue of the rotarod assay is quite modest. Were any other motor phenotypes rescued? It would be nice to see rescue at the neuropathological level.

Statistics

– The statistics for the cell and fly work all employ a 2-tailed Student T-test. However, given that there are at least 3 in dependent groups being compared on each graph, a one-way ANOVA with post hoc test (e.g. Bonferroni) would be better used.

*Reviewer #2 (Recommendations for the authors):*

This article aims to extend human disease-related studies of PLA2G6 from fly models to iPS-neurons, mouse models, to look for drugs that suppress phenotypes and test them, and to attempt AAV whole body rescue. Generally, each of these questions/aims/experiments is excellent, but as presented, it's a bit of an underdeveloped hodgepodge of results, with each experiment somewhat underdeveloped or analyzed for the respective phenotype, in my opinion. I think the general thrust of the experiments is excellent. But the data are relatively cursory in many instances. Further development and characterization of the phenotypes would require quite a bit of work but vastly improve the paper.

1. The phenotypes characterized in iPS neurons are somewhat cursory. First the lipid analyses. GlcCer levels are measured, but I cannot find any description of this in the methods. Also, it would seem important to do thorough lipidomics in these cell types, as well as the mouse brains. Are other sphingolipids or glycosphingolipids affected?

2. Second, lysosomal phenotypes are measured as cells with abnormalities. This needs better quantification, such as lysosomal number and intensity per cell. Also, there is no other phenotyping of the lysosomes for function or lysosomal proteins. This seems to need much more.

3. Third, mitochondria are stained for. These images are really poor and hard to interpret anything with respect to morphology. This also needs some sort of quantification. Later in the paper, quantifying with reference to expression of a rescue vector per cell would help a lot. There is also no functional characterization of mitochondria. This seems to need much more than staining.

4. In mice, the primary phenotypes are RotoRod and survival. These seem okay. However, neuropath of rescue studies with AAV or lentivirus would be good. This would also need some quantification.

5. The phenotyping for the drug rescue studies consists of screening in flies and then lysosomal staining in iPS neurons. This seems inadequate. If the drugs really do work, then an extensive analysis of the ones that do work and some controls would be much more convincing. What happens to the lipids? The lysosomal and mitochondrial morphology and function? The mito morphology shown again seems uninterpretable and not well quantified. Also, did the authors try a rescue with one of these drugs in the mouse model?? That would seem to be the obvious step.

6. Instead, the authors move on to rescuing the mouse model with AAV expressing the enzyme and get the somewhat expected result with replacing the gene in all cell types. What is shown is RotoRod and lifespan and body weight, but what about the lipids, the neuropath defects, etc.?

7. Little is made of the enzymatic function of PLA2G6 and how it relates to the phenotype unfortunately. Also, it would be good to see if the same lipid phenotypes are present in the current systems (cells, mice) and whether they are rescued with the AAV in this system.

---

## [Author Response]

Reviewer #1 (Recommendations for the authors):Points to address:iPSC/NPC work– While the patient IPSC-derived neuronal cell data is interesting and confirms previous findings in *Drosophila* models of PLA2G6 knockout, it does not fully recapitulate all the pathologies previously reported by the authors in the INAD flies. The fly work led to the hypothesis that loss of PLA2G6 leads to loss of normal retromer function with consequent sphingomyelin/CPE mis trafficking and endolysosomal dysfunction, resulting in elevated ceramide levels. Does loss of PLA2G6 in the patient-derived neuronal cells lead to retromer dysfunction and increased ceramide levels? Are levels of Vps35 or Vps26 altered in the patient derived cells?

We thank the reviewer for these questions. To address these questions we tested a few commercially available antibodies that recognize Vps26 or Vps35. We identified a Goat polyclonal antibody (Abcam; ab10099) that detects the endogenous Vps35 punctae in Patient-derived NPCs as well as in mice. By using this antibody, we found that the number of Vps35 punctae is much higher in the genetically corrected NPCs (29-2) than the INAD patient derived NPCs (29-1) (Figure 5A, a-b and B). This is consistent with our finding in flies that PLA2G6 improves the stability of Vps35 (Lin et al., 2018).

Moreover, we tested whether the selected drugs or the expression of human PLA2G6 can restore retromer levels in INAD patient derived NPCs. The addition of Ambroxol, Azoraminde, Desipramine or Genistein to the patient derived NPCs did not restores Vps35 levels (Figure 5A, a-f and B). These data suggest that these drugs may function downstream of the retromer. Ambroxol promotes GCase activity to degrade GlcCer in lysosomes, Azoraminde promotes UPR in the ER, whereas Desipramine blocks ceramide salvage pathway and Genistein enhances lysosome biogenesis. Indeed, these actions are all downstream of the retromer function.

In contrast to the drugs, the expression of the human PLA2G6 restored Vps35 levels (Figure 5A, a,b, g-i and B). This is consistent with our model that PLA2G6 binds to Vps35 and Vps26 to stabilize the retromer complex. In summary, the expression of the human PLA2G6, but not the identified drugs restores Vps35 levels. The expression of the human PLA2G6 promotes retromer levels and function and facilitates the recycling of the proteins and Ceramides to reduce lysosomal expansion and neurodegeneration. The data are now presented in the newly generated Figure 5.

It is interesting that GlcCer levels are up, but it would be useful to look specifically at different ceramide species. The reason for this is that the drugs previously used to rescue neurodegeneration were focused on reducing ceramide levels and not GlcCer. The significance of elevated GlcCer is not clear, and indeed drugs used in Gaucher's disease patients to specifically lower GlcCer were not successful in rescuing the PLAn phenotypes in this study.

To address this question, we performed lipidomic assays to measure the levels of ceramides and their derivatives in the INAD patient-derived NPCs (29-1) and the genetically corrected NPCs (29-2). As shown in Figure 1H, we found that the total Cer and total HexCer levels are about two fold higher in the patient derived NPCs (Figure 1H; left panel and Figure 1—source data 3). However, although the levels of Sphinganine and

Sphingosine are increased in the patient derived NPCs, they are not statistically significantly increased (Figure 1H; right panel and Figure 1—source data 3). These data show that INAD patient derived NPCs exhibit an elevation of the HexCer (including GlcCer) and many other ceramides. This may also explain why a reduction of GlcCer by drugs used in Gaucher's disease patients fail to suppress neurodegenerative phenotypes in the INAD flies. The data are now presented in the manuscript (Figure 1H and Figure 1—source data 3).

– Figure 1E: It is not clear how the mitochondrial fragmentation phenotypes were quantified. How many cell/images were quantified? Was any quantification performed for the LAMP2 immunostaining? Although it looks like there is an obvious difference between patient and control NPCs, it should at least be stated that the images are representative.

To address the reviewer comments, we quantified the length of mitochondria (Figure 1E and 4E) and the number of LAMP2 punctae (Figure 1E). The n number is indicated in the figure legends.

– The current study demonstrates mitochondrial morphological abnormalities (with fragmentation). Is there also evidence of mitochondrial dysfunction in the patient-derived cells?

To address the reviewer comment, we measured ATP levels in 29-1, 29-2 (corrected cells) and 29-3 patient derived NPCs. The ATP levels are higher in the genetically corrected cells (29-2) than in the patient NPCs (29-1 and 29-2). These data indicate that the mitochondrial function is indeed defective in the patient derived NPCs (Figure 1G). We further show that this reduction in ATP levels is mildly but significantly suppressed by treating the NPCs with the four drugs, Ambroxol, Azoraminde, Desipramine or Genistein (Figure 3E), as well as the expression of human PLA2G6 with the gene therapy constructs (Figure 4E).

Mouse work– Again, as with the patient-derived cell data, the mouse work is of significant interest but would benefit from exploration with regards to retromer function, ceramide levels and mitochondrial function in view of the previous hypothesis put forward from the fly work. The authors state that there is ceramide accumulation in the introduction summary and in the summary at the end of the mouse results – strictly speaking accumulation of GlcCer has been shown.

In our original version, we have documented the following phenotypes associated with the *KO/G373R* and or *G373R/G373R* mice: elevated levels of GlcCer (Figure 2C); expansion of lysosomes (Figure 2D); disruption of the mitochondria (Figure 2—figure supplement 1A-C); and presence of MVB and TVS in the Purkinje cells (Figure 2—figure supplement 1B-C).

In the revised manuscript, we performed lipidomic assays to measure the levels of total Cer, total HexCer as well as Sphinganine and Sphingosine in the cerebellum of the *KO/G373R* mice. However, unlike in flies and human cells, we did not observed an obvious change in HexCer as well as other ceramides (Figure 2—figure supplement 1D and Figure 2—source data 1). Given that the GlcCer antibody shows obvious changes that are consistent among the three species, we argue that the lipidomic assays are not sensitive enough to detect the increase in GlcCer because only Purkinje cells seem to be affected. In summary, the changes maybe masked by cells that do not show an obvious increase. We have now included these data in the manuscript.

In summary, we observed elevation of GlcCer, expansion of lysosomes, disruption of mitochondria, reduction of Vps35 punctae, and increased number of MVB and TVS in the Purkinje cells of the *KO/G373R* mice. These phenotypes are consistent with what we observed in the INAD flies and human cells, suggesting that these defects are evolutionary conserved.

Drug screening– The general feel of this work is that although it is of considerable interest, because of the potential for drug-repurposing for therapeutic purposes in PLAN, it has not been performed very rigorously. The candidate compounds have only been tested using one fly readout and a western blot in NPCs. Although statistically significant, the biological relevance of the degree of rescue of bang sensitivity is difficult to assess based on this data alone. This work would benefit from confirmatory studies of the molecular effectiveness of the drug. Ideally it would next be tested in the mouse models to show true efficacy at the lifespan and behavioural phenotype level. At the very least, validation of the drugs that were effective at rescuing bang-sensitivity require further validation in vivo in the fly model (e.g. ERG, photoreceptor loss suppression).

To address the reviewer’s comment, we tested the efficiency of the drugs using other assays. We performed ERG assays and quantified photoreceptor loss in INAD flies. As shown in Figure 3—figure supplement 1A-C, the defective electroretinogram (ERG) phenotypes associated with INAD flies are significantly improved when Ambroxol, Azoraminde, Desipramine or Genistein was added to the food. Moreover, the loss of photoreceptors phenotype is also reduced in INAD flies raised in the food containing each of the four drugs (Supplement Figure 5D-E). In summary, we provide more evidence showing that the four drugs can suppress the loss of PLA2G6-induced phenotypes in flies.

Gene therapy work– While this work is of significant interest, the only cellular defect that is shown to be improved in the patient-derived NPCs is that of the mitochondrial fragmentation and LAMP2 expression. A reduction in ceramide levels or improvement in retromer function would support the efficacy at the cellular level. It is not clear how the fragmentation phenotype has been quantified for the NPCs (as in Figure 1E). The figure legend does not state that the images are representative. There is no indication of how many cells/images were quantified in the figure legend or in the methods.

We thank the reviewer for the suggestions. We have done the following experiments to address the concerns:

1. To address retromer function, we now assess the levels of Vps35 in the treated and untreated patient NPCs. We found that expression of the human PLA2G6 restores Vps35 levels (Figure 5A, a,b, g-i and B).

2. To assess GlcCer levels in mice, we quantified the GlcCer levels in Purkinje cells prior to and after expression of human PLA2G6 using AAV injections. As shown in Figure 7A, these injections strongly suppresses the levels of GlcCer in Purkinje cells based on Ab staining.

3. To quantify the fragmentation phenotype, we now measure the length of mitochondria a shown in Figure 4E. Moreover, we also measured the levels of ATP in treated or untreated NPCs. The data show that expression of human PLA2G6 using the gene therapy construct improves morphology and function of mitochondria.

4. We now added how many images were quantified.

– The rescue of the rotarod assay is quite modest. Were any other motor phenotypes rescued? It would be nice to see rescue at the neuropathological level.

To explore the rescue at the neuropathological level, we scarified two injected KO/G373R mice at ~P300. These two mice were not able to stay on the rod in a rotarod assay. However, they behave relatively normal in their home cage (Supplemental video). As shown in Figure 7A, we found that the GlcCer levels are strongly suppressed in the Purkinje cells of these two P300 animals when compared to the uninjected KO/G373R mice that are 140 days old (Figure 7A). Moreover, the levels of LAMP2 are also reduced in the Purkinje cells of these injected mice (Figure 7B). Hence, the injection of the gene therapy construct restores GlcCer levels and alleviates lysosome expansion.

Next, we assessed the levels of Vps35 in the injected and uninjected KO/G373R mice. Similar to our observations in INAD flies and INAD patient-derived NPCs, the levels of Vps35 is significantly reduced in the KO/G373R mice (Figure 7C). Interestingly, expression of human PLA2G6 not only restores the levels of Vps35, but also enhances its expression (~2 folds of the wild-type mice) (Figure 7C). In summary, expression of human PLA2G6 using our gene therapy construct in KO/G373R mice increases Vps35 punctae. This promotes retromer functions and facilitates the recycling of GlcCer and other cargoes. This in turn reduces lysosomal stress and alleviates neurodegenerative phenotypes. All the data are now included in the manuscript.

Statistics– The statistics for the cell and fly work all employ a 2-tailed Student T-test. However, given that there are at least 3 in dependent groups being compared on each graph, a one-way ANOVA with post hoc test (e.g. Bonferroni) would be better used.

In the figures we only compare each of the treated group to the control. We did not compare all the groups together in one test. To avoid a misunderstanding, we now highlight this point carefully in the figures or in the figure legends.

Reviewer #2 (Recommendations for the authors):This article aims to extend human disease-related studies of PLA2G6 from fly models to iPS-neurons, mouse models, to look for drugs that suppress phenotypes and test them, and to attempt AAV whole body rescue. Generally, each of these questions/aims/experiments is excellent, but as presented, it's a bit of an underdeveloped hodgepodge of results, with each experiment somewhat underdeveloped or analyzed for the respective phenotype, in my opinion. I think the general thrust of the experiments is excellent. But the data are relatively cursory in many instances. Further development and characterization of the phenotypes would require quite a bit of work but vastly improve the paper.1. The phenotypes characterized in iPS neurons are somewhat cursory. First the lipid analyses. GlcCer levels are measured, but I cannot find any description of this in the methods. Also, it would seem important to do thorough lipidomics in these cell types, as well as the mouse brains. Are other sphingolipids or glycosphingolipids affected?

a. The levels of GlcCer in INAD patient cells or mice were detected by Immunostaining using a rabbit polyclonal antibody that specifically recognizes GlcCer. In the original manuscript, we described this method in the Materials and methods section (Line 802-828).

b. To address the ceramide question, we performed lipidomic assays to measure the levels of ceramides and their derivatives in the INAD patient-derived NPCs (29-1) and the genetically corrected NPCs (29-2). As shown in Figure 1H, we found that the total Cer and total HexCer levels are about two fold higher in the patient derived NPCs (Figure 1H; left panel and Figure 1—source data 3). However, although the levels of Sphinganine and Sphingosine are increased in the Patient-derived NPCs, they are not statistically significant (Figure 1H; right panel and Figure 1—source data 3). These data show that INAD patient derived NPCs exhibit an elevation of the HexCer (including GlcCer) and many other ceramides. This may also explain why a reduction of GlcCer by drugs used in Gaucher's disease patients fail to suppress neurodegenerative phenotypes in the INAD flies.

c. In the revised manuscript, we performed lipidomic assays to measure the levels of total Cer, total HexCer as well as Sphinganine and Sphingosine in the cerebellum of the *KO/G373R* mice. However, unlike in flies and human cells, we did not observed an obvious change in HexCer as well as other ceramides (Figure 2—figure supplement 1D and Figure 2—source data 1). Given that the GlcCer antibody shows obvious changes that are consistent among the three species, we argue that the lipidomic assays are not sensitive enough to detect the increase in GlcCer because only Purkinje cells seem to be affected. In summary, the changes maybe masked by cells that do not show an obvious increase. We have now included these data in the manuscript.

2. Second, lysosomal phenotypes are measured as cells with abnormalities. This needs better quantification, such as lysosomal number and intensity per cell. Also, there is no other phenotyping of the lysosomes for function or lysosomal proteins. This seems to need much more.

a. To address the reviewer’s concern, we quantified the average number of lysosome per cell. The data are now presented in Figure 1E, next to the images.

b. In addition to assess the number of lysosomes, we also detected LAMP2 levels in the patient derived cells. These data are in Figure 1F; Figure 3C-D; Figure 4C; and Supplement Figure 1E of the manuscript.

3. Third, mitochondria are stained for. These images are really poor and hard to interpret anything with respect to morphology. This also needs some sort of quantification. Later in the paper, quantifying with reference to expression of a rescue vector per cell would help a lot. There is also no functional characterization of mitochondria. This seems to need much more than staining.

In this revised manuscript, we carefully adjusted the quality of the images. We use arrows to indicate the fragmented and enlarged mitochondria. We use arrowheads to highlight the normal elongated mitochondria (Figure 1E and Figure 4D). We also quantified the length of the mitochondria to provide additional evidence to support our arguments. These data are now presented in Figure 1E and Figure 4E.

To assess the functional of the mitochondria, we measured ATP levels in 29-1, 29-2 and 29-3 patient derived NPCs. We found that the ATP levels are lower in the patient cells (29-2). In summary, our data suggests that the mitochondrial morphology and function are both impaired in the patient derived NPCs (Figure 1G).

4. In mice, the primary phenotypes are RotoRod and survival. These seem okay. However, neuropath of rescue studies with AAV or lentivirus would be good. This would also need some quantification.

To explore the rescue at the neuropathological level, we scarified two injected KO/G373R mice at ~P300. These two mice were not able to stay on the rod in a rotarod assay. However, they behave relatively normal in their home cage (Supplemental video). As shown in Figure 7A, we found that the GlcCer levels are strongly suppressed in the Purkinje cells of these two P300 animals when compared to the uninjected KO/G373R mice that are 140 days old (Figure 7A). Moreover, the levels of LAMP2 are also reduced in the Purkinje cells of these injected mice (Figure 7B). Hence, the injection of the gene therapy construct restores GlcCer levels and alleviates lysosome expansion.

Next, we assessed the levels of Vps35 in the injected and uninjected KO/G373R mice. Similar to our observations in INAD flies and INAD patient-derived NPCs, the levels of Vps35 is significantly reduced in the KO/G373R mice (Figure 7C). Interestingly, expression of human PLA2G6 not only restores the levels of Vps35, but also enhances its expression (~2 folds of the wild-type mice) (Figure 7C). In summary, expression of human PLA2G6 using our gene therapy construct in KO/G373R mice increases Vps35 punctae. This promotes retromer functions and facilitates the recycling of GlcCer and other cargoes. This in turn reduces lysosomal stress and alleviates neurodegenerative phenotypes. All the data are now included in the manuscript.

5. The phenotyping for the drug rescue studies consists of screening in flies and then lysosomal staining in iPS neurons. This seems inadequate. If the drugs really do work, then an extensive analysis of the ones that do work and some controls would be much more convincing. What happens to the lipids? The lysosomal and mitochondrial morphology and function? The mito morphology shown again seems uninterpretable and not well quantified. Also, did the authors try a rescue with one of these drugs in the mouse model?? That would seem to be the obvious step.

We thank the reviewer for the suggestion and performed the requested experiments. We now present the following data to document the rescue effect of the selected drugs in INAD flies and patient derived NPCs.

a. The four drugs, Ambroxol, Azoraminde, Desipramine or Genistein, alleviate bang-sensitivity of the INAD flies (Figure 3B).

b. The drugs restore the ERG defects, including the loss of light coincident receptor potential (LCRP) and on-transient, as well as the loss of photoreceptors observed in the INAD flies (Supplement Figure 5).

c. The drugs reduce the LAMP2 accumulation in INAD patient derived NPCs (Figure 3C-D).

d. The drugs restore ATP levels in INAD patient derived NPCs (Figure 4E).

In summary, we argue that the drugs, Ambroxol, Azoraminde, Desipramine or Genistein, should be used as possible therapeutic methods to treat INAD and PARK14 patients. We agree with the reviewer that a rescue with these drugs in the mouse model would be a key next step to strengthen our argument. However, this will take a long time and will require a significant influx in funding.

6. Instead, the authors move on to rescuing the mouse model with AAV expressing the enzyme and get the somewhat expected result with replacing the gene in all cell types. What is shown is RotoRod and lifespan and body weight, but what about the lipids, the neuropath defects, etc.?

Please refer to our response in question 4.

7. Little is made of the enzymatic function of PLA2G6 and how it relates to the phenotype unfortunately. Also, it would be good to see if the same lipid phenotypes are present in the current systems (cells, mice) and whether they are rescued with the AAV in this system.

We have previously shown that the enzymatically dead PLA2G6 fully rescues the lethality as well as other defects associated with INAD flies (Lin et al. 2018). These data indicate that the enzymatic activity may not play a key role in the pathogenesis of INAD. We now mention this in the introduction (Line 97100).